# Computational design of ultra-robust strain sensors for soft robot perception and autonomy

Haitao Yang[1,2,7], Shuo Ding [3,4,7], Jiahao Wang[2,7], Shuo Sun [5], Ruphan Swaminathan [6], Serene Wen Ling Ng[2], Xinglong Pan[2] & Ghim Wei Ho [2] ✉

Compliant strain sensors are crucial for soft robots' perception and autonomy. However, their deformable bodies and dynamic actuation pose challenges in predictive sensor manufacturing and long-term robustness. This necessitates accurate sensor modelling and well-controlled sensor structural changes under strain. Here, we present a computational sensor design featuring a programmed crack array within micro-crumples strategy. By controlling the user-defined structure, the sensing performance becomes highly tunable and can be accurately modelled by physical models. Moreover, they maintain robust responsiveness under various demanding conditions including noise interruptions (50% strain), intermittent cyclic loadings (100,000 cycles), and dynamic frequencies (0–23 Hz), satisfying soft robots of diverse scaling from macro to micro. Finally, machine intelligence is applied to a sensor-integrated origami robot, enabling robotic trajectory prediction (<4% error) and topographical altitude awareness (<10% error). This strategy holds promise for advancing soft robotic capabilities in exploration, rescue operations, and swarming behaviors in complex environments.

Emerging soft robots, notable for their flexible body deformations and outstanding motion agility[1–4], offer compliant, robust, and safe interactions for dynamic tasks in unstructured environments[5–8]. To enable soft robots to truly respond intelligently to the external world, it is integral to incorporate soft perception components, compliant strain sensors, into the robotic structure to allow real-time sensing of various environmental stimuli as feedback and achieve the kinematic estimation of the robot itself and mapping of surroundings for autonomous navigation[9–13]. However, this automation milestone is hindered by high degree-of-freedom (DOF) body deformations and mutative actuating behaviors of robots[14,15], which complicate the robot kinematics and

pose great design challenges for integrated strain sensors. On one hand, to satisfy the diverse sensing demands of various soft robots with distinct actuation behaviors or dimension scales, it is crucial to fabricate strain sensors with user-designated characteristics (e.g., sensitivity, linear working window)[14,16–18] which generally require different design principles and multiple trial-and-error experiments[19–23]. This time-consuming empirical experimentation becomes tricky when there is a need for custom production that is predictive in nature to streamline the design process and minimize iterations. An alternative approach involves the development of sensor modelling tools, utilizing mathematical statistics or physical simulation, to virtually confirm

[1]Institute of Flexible Electronics (IFE) & Frontiers Science Center for Flexible Electronics, Northwestern Polytechnical University, Xi'an, Shaanxi 710072, China. [2]Department of Electrical and Computer Engineering, National University of Singapore, Singapore, 4 Engineering Drive 3, Singapore 117583, Singapore. [3]College of Mechanical and Electrical Engineering, Nanjing University of Aeronautics and Astronautics, Nanjing 210016, China. [4]Department of Biomedical Engineering, National University of Singapore, Singapore, 4 Engineering Drive 3, Singapore 117583, Singapore. [5]Department of Mechanical Engineering, National University of Singapore, Singapore, 9 Engineering Drive 1, Singapore 117575, Singapore. [6]Department of Computer Science, Columbia University, New York, NY 10027, USA. [7]These authors contributed equally: Haitao Yang, Shuo Ding, Jiahao Wang. ✉e-mail: elehgw@nus.edu.sg

and optimize sensor characteristics based on device composition and morphology[24]. However, this approach is challenged by the unpredictable material and structure evolutions of the dynamics of conventional soft strain sensors[15,25].

On the other hand, sensor stability is central to soft robotic automation with accurate closed-loop control, but it is often neglected in academic research[26,27]. This oversight directly impacts the ability of soft robots to accurately estimate themselves and adapt to environmental changes in real-time without the need for recalibration and error correction, periodic resting for materials recovery or energy dissipation, etc[14]. For the state-of-the-art soft strain sensors, their stability tests were mostly performed under monotonic, repetitive, and controlled conditions[16–22,28–35], but do not adequately represent the complexity and uncertainty present in real-world working situations of soft robots. Often, soft robots face external mechanical interruptions or unexpected deformations in noisy environments and need to work intermittently to maintain their performance and reduce the risk of premature failure[2,8,36,37]. In addition, the evolving working speeds (0.1–20 Hz)[8,38–40] of soft robots necessitate that the integrated sensor demonstrates a stable response even under varying frequencies of operation. However, when monitoring these noisy, intermittent, and dynamic robot motions, the integrated strain sensors usually experience material/structure failures, resulting in large signal distortions and degraded feedback[17,22,41–45]. It is significant to develop highly robust sensors that can sustain complex and dynamic operating environments to bridge the gap between soft sensors and robot practicality.

In this work, based on the deterministic crack propagation mechanism, a computational strain sensor design is developed to address both the sensor modelling and sensor stability challenges to realize autonomous soft robot navigation. Firstly, sensor modelling is achieved via precise sensor manufacturing as well as prescribed structure evolution. Basically, through laser-aided fabrication, user-defined interdigital crack arrays were programmed within the micro-crumples of piezoresistive strain sensors, illustrating highly controllable crack propagation behaviors and tunable sensor characteristics. By inputting the sensor structure parameters including crack density and micro-crumple feature, corresponding finite element analysis (FEA) models were established to conduct dual physical fields including mechanical and electrical evolutions, and simulate the sensing curves of different sensors with high accuracy. Secondly, excellent sensor robustness was materialized by the deterministic crack propagation mode and micro-crumple feature. In particular, the as-fabricated sensors showed long-term mechanical robustness under noise interruptions (up to 50% strain), intermittent cyclic loadings (100,000 cycles), and dynamic operation frequencies (0–23 Hz), satisfying the diverse sensing requirements of soft robots from macro to micro scales. With the aid of machine learning algorithms, a sensor-integrated origami robot could realize autonomous robot navigation of high accuracy, including self-estimation (robotic trajectory prediction with <4% error) and surroundings mapping capabilities (topography altitude awareness with <10% error). The convergence of both hardware sensor and software system advancements allows the soft robot to sense and perceive reliably, make informed decisions, and navigate autonomously in complex environments.

## Results

### Computation-guided PCAM sensor design
In this study, we employed environmentally stable single-walled carbon nanotubes (SWNT) to fabricate piezoresistive strain sensor. A two-stage design was developed, incorporating the feature of "programmed cracks array within micro-crumples," resulting in what we abbreviated as PCAM sensor. First, as shown in Fig. 1a, a computer-programmed laser machine creates user-defined interdigital patterns that consist of an interlocking comb-shaped crack array on SWNT-coated polystyrene (PS) films (SWNT thickness was ca. 600 nm, see

details in Supplementary Figs. 1–4, Methods, and Supplementary Note 1). The laser beam power was precisely controlled at 0.08 mW to ensure only the upper SWNT layer along with the interdigital lines was trimmed to form the cracks array while the PS substrate was kept intact (Supplementary Fig. 5). The width of the programmed crack was ca. 20 μm (Fig. 1b).

Second, the laser-processed crack array underwent thermally induced dimensional shrinkage. The PS substrate, being thermally responsive, contracts in a biaxial direction above the glass transition temperature ($T_g$) (ca. 100 °C). By harnessing the surface instability of the SWNT-coated PS devices at 140 °C, as shown in Fig. 1c and Supplementary Fig. 6, the upper SWNT layer was biaxially compressed into isotropic crumples, leading to the overlaying of the laser-programmed cracks. Afterward, the shrunken device was top coated with a 2 mm thick Ecoflex (00–35), and the rigid PS substrate was removed to obtain an Ecoflex-coated SWNT device, defined as the PCAM sensor (see fabrication details in Methods).

FEA and in-situ scanning electron microscope (SEM) studies were conducted to explore the structural evolution of the PCAM sensor under uniaxial strains. As shown in Fig. 1d and Supplementary Fig. 7, before stretching, a uniform stress distribution within the SWNT layer without crack was observed. With increasing applied strains to 60%, localized stress along the programmed interdigital pattern emerged, leading to the growth of interdigitated cracks. As a result, the length of the conductive pathway gradually increased, which induced higher in-plane resistances that serve as a strain sensor. Figure 1e presents the strain sensing profiles of a PCAM sensor with a simple interdigital pattern 1 (see dimension details in Supplementary Fig. 4b), showing a three-stage sensing response. First, there is a silent region from 0 to ca. 20% strains, where the resistance changes were small. Afterward, it experienced a quick response from 20% to ca. 60% and then gradually reached a plateau. We implemented in situ SEM studies and FEA simulation to reveal the structure evolutions and mechanism. As shown in Supplementary Fig. 7, there was relative uniform stress distribution across the SWNT sensing layer when applying <20% strain, and the in-plane micro-crumples were gradually deformed to release the compressive pre-strains from thermal contraction during sensor fabrication, and slowly widening the cracks. When the applied strain ≥20%, low stress was concentrated on the crack trace locations, which were more easily deformed, and the cracks started growing quickly until ca. 60% strain. Thereafter, the length of each crack reached its maximum value, and further cracks grew in the width instead of the length, minimally affecting the conductive pathways within the SWNT layer of sensor. As a result, the sensing signal changes became milder at this stage.

Sensor fabrication reproducibility is highly critical for practical applications to ensure minimal device-to-device error and avoid the need for frequent re-calibration. In this study, as shown in Fig. 1e, the PCAM sensor reproducibility was accessed based on three sensor replicates, all of which exhibited highly consistent sensing curves with small signal variations.

Two characteristics of a strain sensor are generally evaluated, including sensitivity and linear working window. The sensitivity of a strain sensor is characterized by gauge factor (GF), as defined in Eqs. 1 and 2,

$$\delta_\varepsilon = \frac{R_\varepsilon - R_0}{R_0} \tag{1}$$

$$GF = \frac{\delta_\varepsilon}{\varepsilon} \tag{2}$$

where $\delta_\varepsilon$ is the relative resistance change at $\varepsilon$ strain, $\varepsilon$ denotes the applied strain, $R_0$ and $R_\varepsilon$ represent the initial resistance and the resistance under $\varepsilon$ strain, respectively. The linear working window of a

 

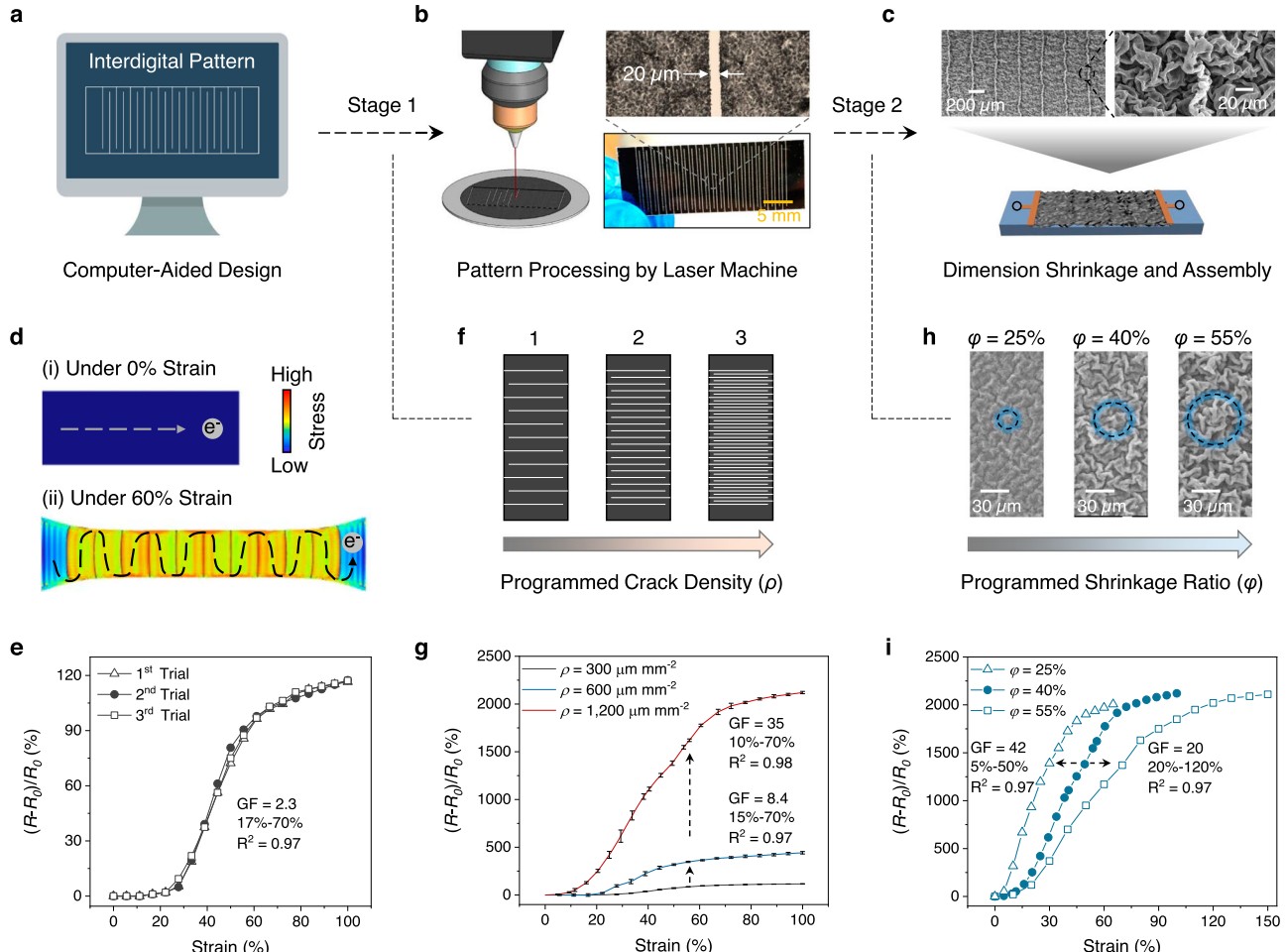

**Fig. 1 | Computation-guided PCAM sensor design. a** Computer-aided design of interdigital crack pattern. **b** Illustration of pattern processing by a laser machine and optical images of the processed crack array. **c** SEM images of the fabricated strain sensor. **d** FEA simulated stress maps of PCAM sensor under strains. **e** Strain sensing profiles of a PCMA sensor ($\rho$ = 300 $\mu$m mm$^{-2}$, $\varphi$ = 40%) with three replicas. **f** Interdigital crack pattern with different crack densities ($\rho$ values). **g** Strain sensing profile of PCAM sensors with different $\rho$ values. The $\varphi$ values of all sensors were kept as 40%. The error bars were calculated based on three sensor replicas. **h** SEM images of PCAM sensors with different $\varphi$ values. **i** Strain sensing profile of PCAM sensors with different $\varphi$ values. The $\rho$ values of all sensors were kept at 1200 $\mu$m mm$^{-2}$.

strain sensor is determined by the strain range where its resistance increased linearly with the applied strain.

Notably, the GF and linear working window of a PCAM sensor can be tuned in a programmable fashion by setting the fabrication parameters in the two design stages. Particularly, GF values are adjustable by varying crack densities ($\rho$) in the first design stage. The $\rho$ is defined by Eq. 3,

$$\rho = \frac{\sum(L_{crack})}{S_{area}} \quad (3)$$

where $\sum(L_{crack})$ is the cumulative length of surface cracks, and $S_{area}$ is the surface area of the SWNT layer. When adopting crack arrays 1, 2, and 3 in Fig. 1f (see dimension details in Supplementary Fig. 4), the $\rho$ values were calculated as 300, 600, and 1200 $\mu$m mm$^{-2}$, respectively. With increasing $\rho$ values, as shown in Fig. 1g, PCAM sensors showed a little increased range of linear working windows from 17–70% to 15–70% and 10–70%, while their GF was greatly improved from 2.3 to 8.4 and 35. Sensor reproducibility at higher $\rho$ values were also tested and discussed in Supplementary Note 2. Further improvement of the sensor's GF (>200) was achievable by controlling the laser etching depth of the SWNT layer (see Supplementary Fig. 8 and Supplementary Note 3 for discussion).

In addition to GF, the sensor's linear working windows were determined during the second-stage design. As shown in Supplementary Fig. 6, by setting different heating durations, the shrinkage ratio ($\varphi$) of the PS film was tunable from 0 to 55%. The $\varphi$ was defined and calculated by Eq. 4,

$$\varphi = \frac{D_0 - D_{After}}{D_0} \quad (4)$$

where $D_0$ and $D_{after}$ are dimensions of the PS shrink film before and after thermal contraction. As shown in Fig. 1h, i, when the $\varphi$ values were controlled as 25%, 40%, and 55%, the resulting SWNT layer showed increasing crumples sizes with hierarchical structures, and the corresponding PCAM sensors demonstrated a shifting linear working window from 5–50% to 10–70% and 20–120%. As both parameters of $\rho$ and $\varphi$ could be computationally determined before sensor fabrication, it signifies a computation-guided sensor design.

## Physical modelling of the PCAM sensor

The computational designed sensor provides a unique opportunity for the physical modelling of the developed PCAM sensors to predetermine their sensing performances without conducting experiments. As shown in Fig. 2a, an FEA model consisting of a SWNT layer

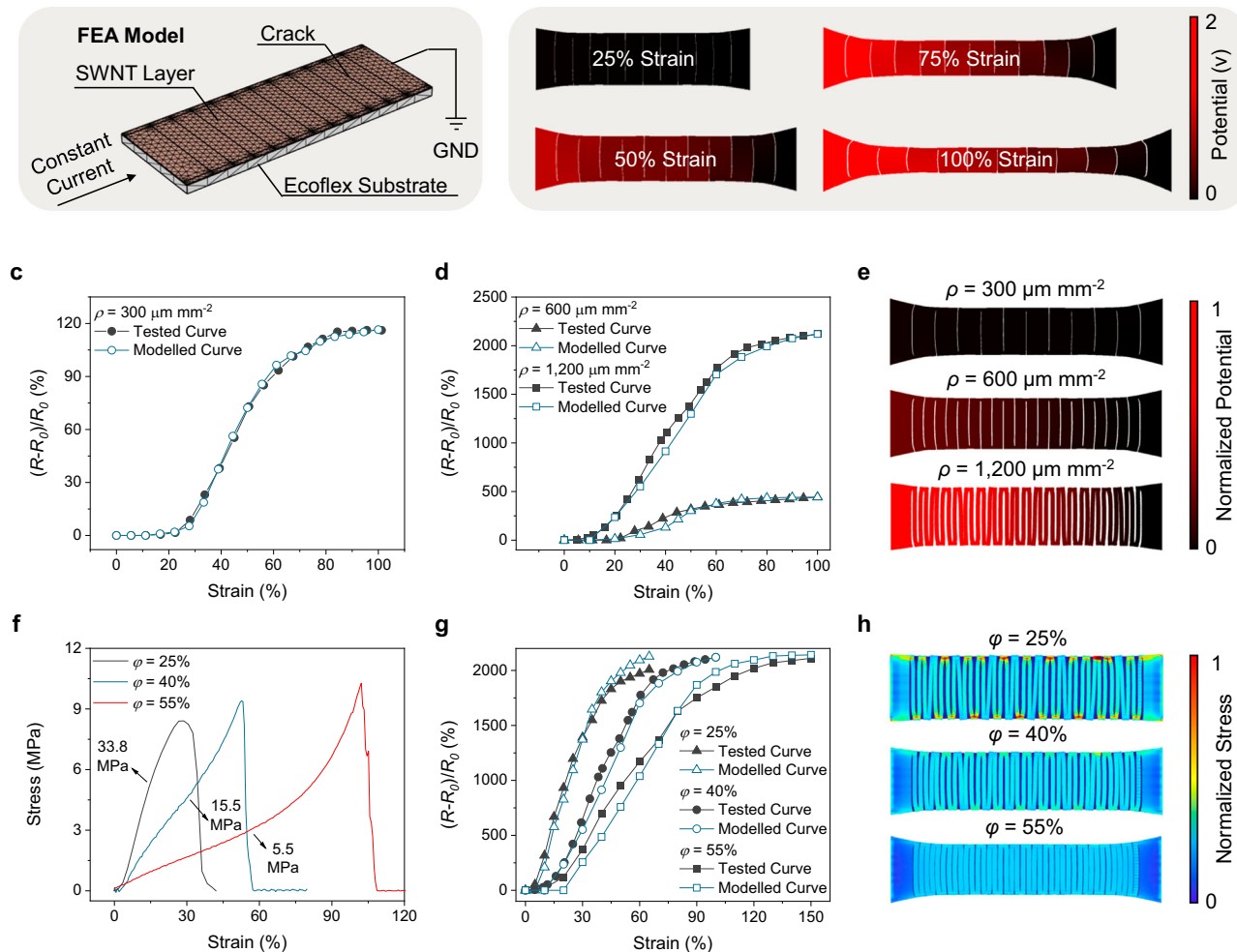

**Fig. 2 | Physical modelling of the PCAM sensor. a** FEA model of a PCAM sensor ($\rho = 300\,\mu m\,mm^{-2}$; $\varphi = 40\%$). GND refers to ground. **b** Electrical evolutions of the PCAM sensor ($\rho = 300\,\mu m\,mm^{-2}$; $\varphi = 40\%$) under strains. **c** Experimental and modelled sensing curves of a PCAM sensor ($\rho = 300\,\mu m\,mm^{-2}$; $\varphi = 40\%$). **d** Experimental and modelled sensing curves of PCAM sensors with different $\rho$ values ($\varphi = 40\%$).

**e** Simulated potential drops of PCAM sensors under 50% uniaxial strains. ($\varphi = 40\%$). **f** Stress-strain curves of SWNT layers with different $\varphi$ values ($\rho = 1200\,\mu m\,mm^{-2}$). **g** Experimental and modelled sensing curves of PCAM sensors with different $\varphi$ values ($\rho = 1200\,\mu m\,mm^{-2}$). **h** Simulated stress distribution maps of PCAM sensors under 50% uniaxial strains. ($\rho = 1200\,\mu m\,mm^{-2}$).

and an Ecoflex substrate was constructed to simulate the bilayer sensor structure, where $\rho$ and $\varphi$ were input as model structure parameters (see details in Methods). With the parameter setup and the application of a constant current in the FEA model, a dual physical field including mechanical and electrical simulation was simultaneously implemented to extract the sensor model's resistance evolution under strains. Figure 2b and Supplementary Movie 1 illustrate the dynamics of a PCAM sensor ($\rho = 300\,\mu m\,mm^{-2}$; $\varphi = 40\%$). As the PCAM sensor was subjected to uniaxial strain loading, its surface cracks gradually grew from 0% to 100% strain. Meanwhile, the surface potential drops gradually enlarged from 25% to 75% strains but stabilized afterward until 100% strains. By extracting the potential change as well as the current data, the relative resistance change profile of the specific sensor model was calculated and plotted in Fig. 2c, which is consistent with the experimental results. Following similar procedures, in Fig. 2d, PCAM sensors with $\rho$ as 600 and 1200 $\mu m\,mm^{-2}$ were also modelled and the results showed good agreement with the experimental data. It is worth to note that current sensor performance modelling was made during strain loading process, yet the strain unloading process remained unsolved due to the difficulties in simulating the spontaneously stress relaxation of elastic polymer segments of sensor substrate which is a nanoscale physical process and shows non-steady feature. The FEA tool not only

facilitates experiment-free sensing performance modelling, but also provides inverse insights into the sensor design principles. For example, the electrical distribution maps in Fig. 2e delineate the relationship between GF and $\rho$. As such, a larger $\rho$ leads to a higher surface potential drop under the same strain, which induces an increasing relative resistance change of the sensor to attain a higher GF.

The PCAM sensor modelling with varying $\varphi$ was also investigated. Firstly, in Fig. 2f, the stress-strain curves of the SWNT layers with different $\varphi$ values were measured by a tensile test machine, and the curve slope represents the Young's modulus of SWNT layers. According to the results, when $\varphi$ increases to 25%, 40%, 55%, the Young's modulus of SWNT layer showed a decreasing trend from 33.8 to 5.5 MPa. Lower Young's modulus means the deformation of SWNT layer require smaller force. Such a trend is attributed to the different sizes of micro-crumples within the SWNT layer. As shown in Supplementary Fig. 9, micro-crumples were generated after the dimension shrinkage during the sensor fabrication. This kind of micro-structures stored elastic force within the SWNT layer. With higher $\varphi$ values, the size of micro-crumples was enlarged, corresponding to a higher stored elastic force like a more compressed spring. Therefore, SWNT layer with higher $\varphi$ value requires a smaller force to induce the deformation, which is reasonable to illustrate a lower Young's modulus. By inputting Young's

modulus data, the dynamics of PCAM sensors under varying $\varphi$ were modelled and their sensing profiles were extracted in Fig. 2g, which corroborates the experimental results. Furthermore, these FEA models explain the phenomenon of working window shifts under different $\varphi$. Figure 2h compares the stress maps of different sensors under 50% strain, demonstrating that the sensor with a higher $\varphi$ exhibits a lower in-plane stress concentration and slower surface cracks propagation. Considering this, it is reasonable to expect a high $\varphi$ to translate into a larger sensor working window.

### Mechanical stability of the PCAM sensor

The mechanical stability of a sensor is crucial for its practical usage in complex environments, which may involve noise interruptions, long-term intermittent cyclic loadings, and dynamic operation frequencies. Herein, the PCAM sensor with a $\rho$ of 1200 mm $\mu$m$^{-2}$ and a $\varphi$ of 40% was selected as a representative case to study, due to its balanced characteristics of GF and linear working window (Fig. 1i). We first conducted some general mechanical testings namely sensing performances of the PCAM sensor under different uniaxial strains (Supplementary Fig. 10) and cyclic bending/twisting loading (Supplementary Fig. 11). In addition, the hysteresis of a PCAM sensor ($U_{hysteresis}$) was quantified by measuring the maximal signal difference

between the stretching and releasing processes, as defined in Eq. 5,

$$U_{hysteresis} = Max \left| \delta_{stretching} - \delta_{releasing} \right| \tag{5}$$

where $\delta_{stretching}$ is the relative resistance change signal, $(R_\varepsilon - R_O)/R_O$, of the PCAM sensor at $\varepsilon$ strain during the stretching process, and $\delta_{releasing}$ is the relative resistance change of the PCAM sensor at $\varepsilon$ strain during the relaxation process. Based on Supplementary Fig. 12, the hysteresis values of PCAM sensor under increasing applied strains (20–70%) were calculated as 0.51, 1.75, 1.91, 2.90, 3.54, 3.63, respectively, as the hysteresis of the Ecoflex substrate increased with the applied strains[46]. Supplementary Note 4 discussed a sandwiched structure to further decrease the sensor hysteresis.

Sensor stability under interrupted mechanical deformations was investigated (Fig. 3a and Supplementary Movie 2), where the sensor experienced a dynamic mechanical loading sequence consisting of stretching, twisting, stretching, bending, and stretching. The PCAM sensor remained stable with consistent sensing signals throughout the multiple stretching processes. On the contrary, a planar sensor (without micro-features) showed large signal fluctuations during the test (Supplementary Movie 3). The exceptional stability of the PCAM sensor was attributed to its micro-crumples that could counteract

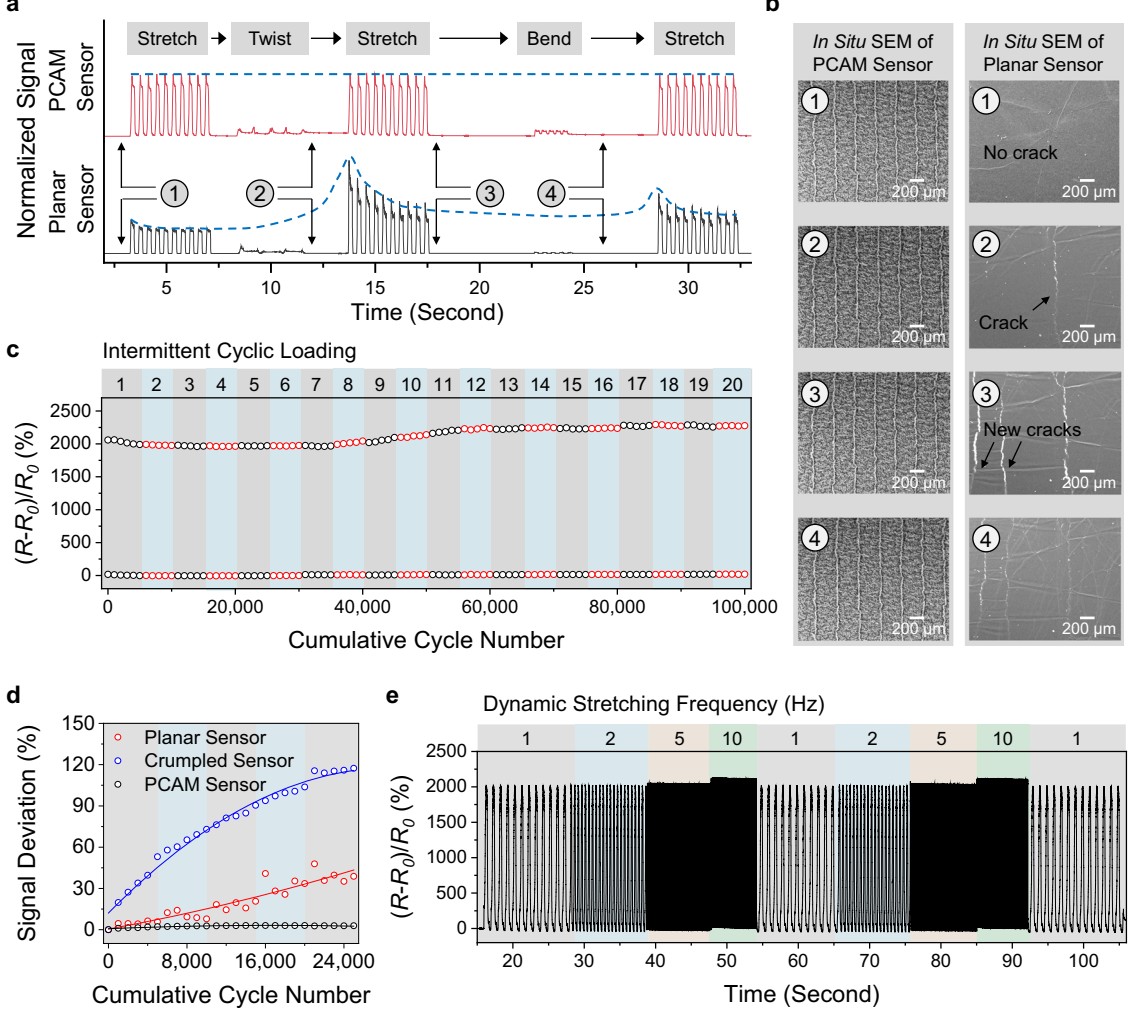

**Fig. 3 | Mechanical stability of the PCAM sensor. a** Sensing profiles of PCAM and planar sensors under a dynamic mechanical loading sequence. **b** In situ SEM images of PCAM and planar sensors during a dynamic mechanical loading sequence. **c** Sensing profiles of a PCAM sensor under intermittent cyclic loading. There were 20 rounds of intermittent cyclic loading. Each cyclic loading consisted of 5000 cycles of uniaxial stretching at 85% strains, with a 1-h time interval between each cycle. **d** Comparison of signal deviations of planar, crumpled, and PCAM sensors under intermittent cyclic loading. Herein, the signal deviations were calculated based on Eq. 6. **e** Sensing profile of PCAM sensor under a dynamic stretching frequency range from 1–10 Hz.

multiaxial mechanical impacts without damaging the sensor. This can be inferred from the observation of an intact PCAM sensor structure (Fig. 3b and Supplementary Fig. 13) as opposed to the planar sensor that showed new surface cracks (Supplementary Fig. 14). As discussed in Supplementary Note 5, the tolerance of mechanical noise for the PCAM sensor was ca. 50% strain.

To simulate the intermittent working scenario, the PCAM sensor was subjected to 20 rounds of intermittent cyclic loading. Each cyclic loading consisted of 5000 cycles of uniaxial stretching at 85% strains, with a 1-h time interval between each cycle. The PCAM sensor exhibited robust sensing signals throughout the cumulative 100,000 intermittent cyclic loadings (Fig. 3c and Supplementary Figs. 15, 16). Both the signal baseline and peaks remained consistent, indicating durability and stable sensor performance. For comparison, a planar sensor without micro-features and a crumpled sensor featuring micro-crumples without programmed cracks were fabricated and tested. The corresponding results were recorded in Supplementary Figs. 17 and 18, where both sensors demonstrated significant signal deviations. Herein, sensor signal deviation was defined and quantified by Eq. 6,

$$SignalDeviation^i = \frac{\delta^i - \delta^1}{A} \tag{6}$$

where $\delta^i$ is the sensor's relative resistance change (see definition in Eq. 1) of the $i^{th}$ cycle, $\delta^1$ is the sensor's relative resistance change of 1st cycle, and $A$ is the $\delta$ signal amplitude of the 1st cycle. A higher signal deviation indicates worse sensor stability, and vice versa. As summarized in Fig. 3d, the PCAM sensor maintained a small signal deviation of approximately 6% over the course of 25,000 intermittent cyclic loadings, while the planar and crumpled sensors exhibited much higher signal deviations, which steadily increased to 40% and 120%, respectively.

Additionally, mechanical tests were conducted at varying sensor operation frequencies. First, the PCAM sensor produced steady signals under a static stretching state (i.e., 0 Hz) for over 30 min (Supplementary Fig. 19). Furthermore, the PCAM sensor performances under dynamic stretching frequencies were evaluated using a stretching machine (Supplementary Fig. 20), where the sensor signals remained stable across 1 to 10 Hz (Fig. 3e and Supplementary Movie 4). A slight signal shift at 10 Hz can be attributed to the hysteresis effects of the Ecoflex substrate. Furthermore, the excellent sensor stability of the PCAM sensor was demonstrated under an extremely high working frequency of 23 Hz (Supplementary Fig. 21 and Supplementary Movie 5). Given that state-of-the-art soft actuators operate within a range of mutative speeds (0.1–20 Hz)[8,38–40], it is essential for PCAM sensors to exhibit a stable sensing response across dynamic operation frequencies, making them deployable for a broad range of soft robot applications.

## PCAM sensor-integrated soft robots across scales
These highly robust PCAM sensors were further integrated into a series of soft robots across various scales to enrich robotic sensing capability. The first application of these PCAM sensors was demonstrated by integrating them into a 15 cm-length origami robot (Fig. 4a). Under magnet actuation (see details in Supplementary Note 6), the origami robot showcased multimodal locomotion, including forward/backward movement and left/right turns (Fig. 4b and Supplementary Movie 6), while the integrated body sensors provided real-time electronic feedback. Figure 4c depicts the continuous monitoring of the origami robot during comprehensive navigation involving moving forward 60 steps, moving backward 5 steps, turning left 10 steps, and encountering an obstacle. The on-body sensing profiles were compared to identify the robot actuation states (see analyses in Supplementary Note 7). In addition, according to the sensor signal changes shown in Supplementary Figs. 22–24, the origami robot could also distinguish different surface roughness (defined by the arithmetical

mean height ($R_a$), see Methods), such as the desktop ($R_a = 2.4 \, \mu m$), printing paper ($R_a = 3.5 \, \mu m$), watery surface ($R_a = 0.6 \, \mu m$), or when encountering an obstacle.

Besides, the PCAM sensors were embedded into a pneumatic robot, as shown in Fig. 4d and Supplementary Fig. 25. Soft pneumatic bodies typically exhibit high DOF deformations during actuation, making long-term body sensing a challenging task. However, the SEM image in Fig. 4d showed that the PCAM sensor structure remained integral after repeated pneumatic deformations, enabling the robot sensing of multimodal locomotion, surface identification, and obstacle detection (Supplementary Figs. 26–28). Long-term sensing scenarios were also examined. Based on the setups in Supplementary Figs. 29–32 and Supplementary Note 8, a sensor-integrated pneumatic robot was controlled to complete long trajectories in an artificial terrain (trajectory 1: 25 cm, turning mode, 131 steps; trajectory 2: 60 cm, crawling mode, 221 steps.). Both trajectories were repeated 5 times. The corresponding sensing profiles were recorded (Fig. 4e, f, Supplementary Figs. 33–34, and Supplementary Movies 7 and 8), wherein notable consistency in the sensor signal was observed across all five iterations of each trajectory. Specifically, according to Supplementary Figs. 35–36, the sensor signal deviations (defined in Eq. 6) between the first and fifth iterations of both trajectories were approximately 6%. In contrast, when planar and crumpled sensors were integrated into the pneumatic robot, significant shifts in robot signals were detected after repeated navigations, with peak signal deviations exceeding 50% (Supplementary Figs. 35 and 36). This comparison is summarized in Supplementary Figs. 37 and 38.

Beyond macro-scale robots, we also demonstrated the seamless integration of PCAM sensors with micromachines. Figure 4g illustrates the scalable fabrication of a cross-shaped interdigital pattern with a SWNT film using a laser machine. The crack pattern was cut out, subjected to thermal shrinkage, and integrated with an (Nd-Fe-B)-Ecoflex substrate to form a tetrapod microrobot (see details in Supplementary Fig. 39 and Methods). The dimension of the as-fabricated microrobot was around 700 $\mu m$ (Fig. 4h). After polarization under a strong magnetic field (exceeding 1 T, see setups in Supplementary Fig. 39), the tetrapod microrobot was able to undergo reversible body transformation (Fig. 4i), along with on-body sensing (Supplementary Fig. 40) under repeated magnetic actuation.

## Intelligent sensor network for robotic trajectory prediction
Given the direct correlation between on-body sensor signals and robotic actuation states, an intelligent sensor network was constructed to predict robotic trajectories by machine learning (ML) algorithm. In this work, origami robot was selected as the ML demonstration from three kinds of developed robots, due to its large number of locomotion modes as well as untethered actuation features (see discussion in Supplementary Note 9). As shown in Supplementary Figs. 41 and 42, a sensor-integrated robot was developed, consisting of an origami body, four PCAM sensors, and a magnet actuation platform. Under repeated forward/backward or left/right movements, the robot was able to crawl long distances (>110 cm) with a small system error (relative error < 1%, absolute error < 0.8 cm, see characterizations in Supplementary Fig. 43 and Supplementary Note 10). With the controlled actuation and reliable sensors, an ML-enabled trajectory prediction model was built in two steps. First, as shown in Fig. 5a, b, the sensor-integrated origami robot was placed in an artificial terrain to execute multi-directional trajectories, and the real-time sensor data was collected in a multi-channelled fashion. These sensor data combined with robot actuation information, served as model inputs. Herein, as shown in Fig. 5a, the actuation information contained the instructions from robot control station including crawling direction (1 refers to crawling forward, −1 refers to crawling backward, and 0 refers to no crawling motion) and turning direction (1 refers to turning left, −1 refers to turning right, and 0 refers to no turning motion). Concurrently, the

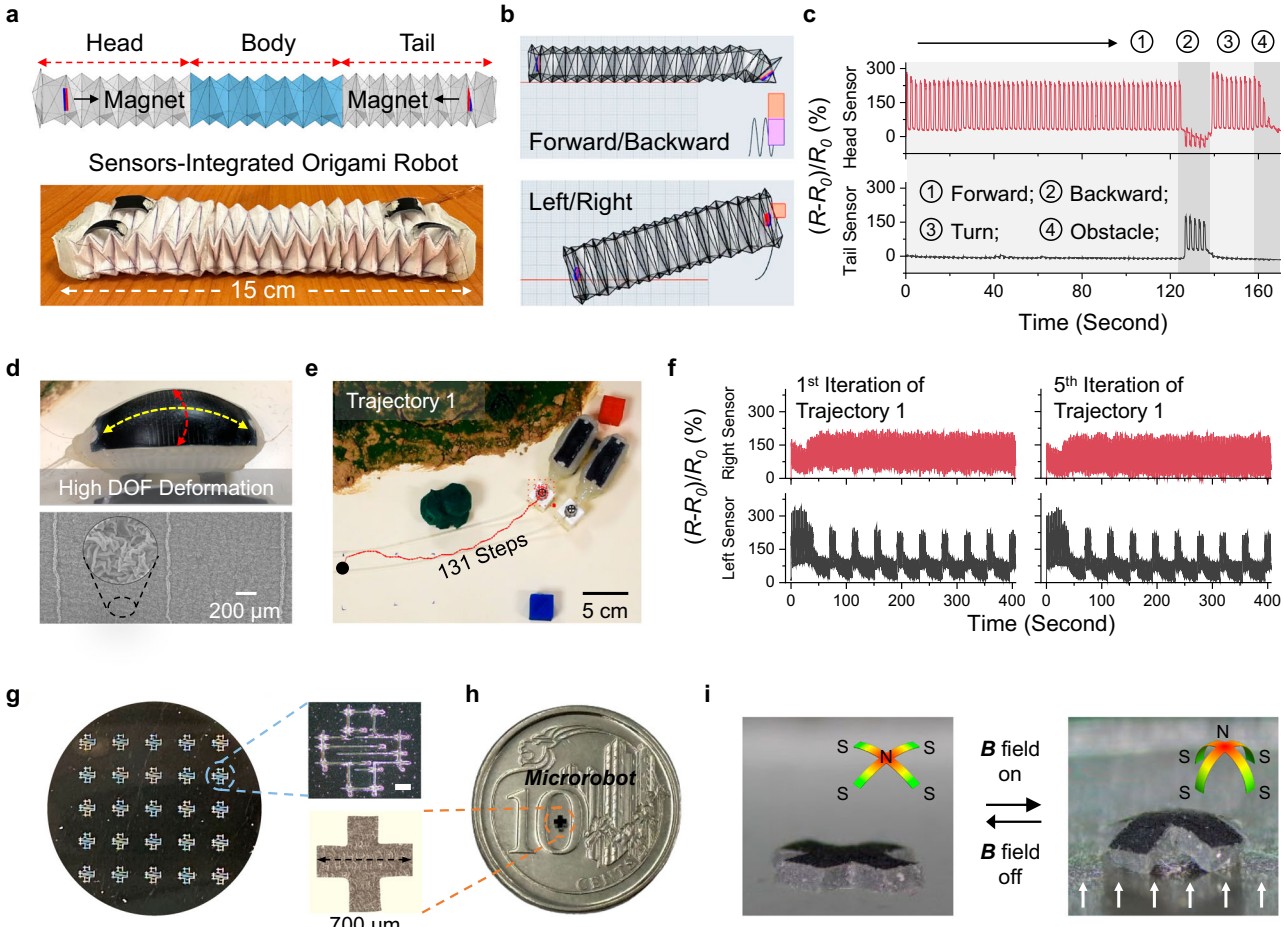

**Fig. 4 | PCAM sensor-integrated soft robots across scales. a** Schematic illustration and digital photo of a sensor-integrated origami robot. **b** Multi-model locomotion of origami robot. **c** Sensing profiles of a sensor-integrated origami robot during comprehensive navigation involving moving forward 60 steps, moving backward 5 steps, turning left 10 steps, and encountering an obstacle. **d** Digital photo of the soft pneumatic robot under gas pressure and SEM photo of the integrated PCAM sensor after repeated pneumatic deformations. **e** Record of trajectory 1 for the soft pneumatic robot under actuation. **f** Sensing profile of PCAM sensor-integrated soft pneumatic robot under 1st and 5th iterations of trajectory 1 (as recorded in **e**), the integrated PCAM sensors showed good signal stability with small signal deviations. **g** Scalable fabrication of a cross-shaped interdigital pattern within an SWNT film by laser machine. Scale bar in the insert photo: 0.4 mm. **h** Digital photo and SEM image of the fabricated tetrapod microrobot. The microrobot dimension was around 700 μm. **i** Tetrapod microrobot was able to undergo reversible body transformation under repeated magnetic actuation.

robot trajectory was recorded by a camera system (see Supplementary Fig. 44), and real-time robot locations in the terrain were extracted using a "Tracker" program (an open-sourced software, see Methods) to use as model outputs. As illustrated in Fig. 5c, the sensing and actuation information were input as the training data to an ML model based on artificial neural network (ANN) for predicting robot trajectories. Detailed ML framework is provided in Methods.

In this study, 38 collected data sets were used as training data to establish the trajectory prediction model (see data files in GitHub). Thereafter, as shown in Fig. 5d, 5 additional data sets, which were never previously presented to the prediction model, were used to evaluate the model prediction accuracy. The relative error (RE) and the absolute error (AE) between the predicted robot location and actual location values were calculated in Eqs. 7 and 8,

$$RE_i = \frac{\sqrt{\left(x_{pre} - x_{real}\right)^2 + \left(y_{pre} - y_{real}\right)^2}}{S} \quad (7)$$

$$AE_i = \sqrt{\left(x_{pre} - x_{real}\right)^2 + \left(y_{pre} - y_{real}\right)^2} \quad (8)$$

where i is the index of the test set (i = 1–5) $x_{pre}$ and $x_{real}$ are the predicted and real robot destination location in the x direction (see the setup of the coordinate system in Fig. 5d) of the $i^{th}$ test set, respectively; $y_{pre}$ and $y_{real}$ are the predicted and real robot destination locations in the y direction of the $i^{th}$ test set, respectively; S is the cumulative robot displacement of the $i^{th}$ test set. After calculation, the model prediction results were displayed in Fig. 5e, f, and Supplementary Fig. 45, where the ANN-predicted robot location successfully tracked the real robot trajectory with high precision. As summarized in Fig. 5g, the RE and AE values among the five test sets were less than 4% and 3 cm, respectively. In addition, to establish a benchmark, another ANN model was trained by using only robot actuation information as the training data. According to the results in Supplementary Fig. 46 and Fig. 5h, without on-body sensor profiles, the benchmark model demonstrated a significantly poorer prediction with 10 times higher RE and AE.

## Surrounding awareness robot navigation

With the assistance of ML, the sensor-integrated robot was also capable of navigating with surrounding awareness. As shown in Fig. 6a, a series of 3D-printed hills was introduced within the artificial terrain to mimic a varying terrain environment. By using the Grasshopper

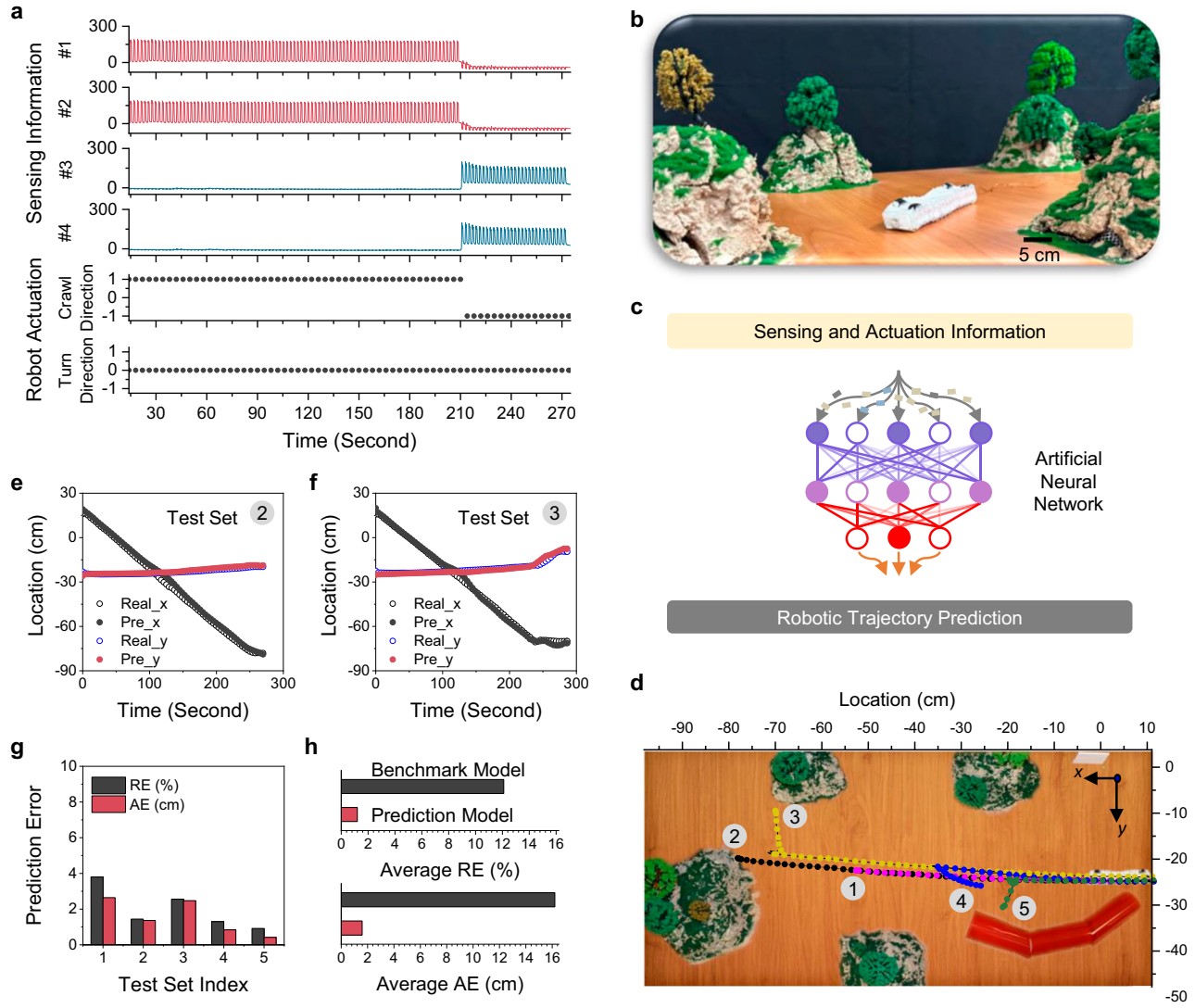

**Fig. 5 | Intelligent sensor network for robotic trajectory prediction. a** Multi-channeled sensing and actuation information of origami robot during navigation. Note: #1, #2, #3, and #4 are sensor location indexes that are shown in Supplementary Fig. 41; For crawling direction: 1 refers to crawling forward, −1 refers to crawling backward, and 0 refers to no crawling motion; For turning direction: 1 refers to turning left, −1 refers to turning right, and 0 refers to no turning motion; (**b**) Digital photo of the sensor-integrated origami robot in artificial terrain. **c** Workflow of the ANN model. **d** Records of robot trajectories from five test sets. **e** Prediction performance of the ANN model on 2nd test set. **f** Prediction performance of the ANN model on 3rd test set. **g** Summary of RE and AE of trajectory prediction model on 5 test sets. **h** Model accuracy comparison between the trajectory prediction model and a benchmark model which only adopted actuation information as the training data. Note: average RE and AE values are calculated based on the prediction results on 3rd, 4th, and 5th test sets.

software, Fig. 6b, c, and Supplementary Movie 9 simulated the origami robot movements when encountering a hill. The changes in distance between two specified origami folds at the robot head were tracked (see details in Methods). According to the result in Fig. 6c (iv), compared to the moving forward mode, the change in distance between the tracked folds during the climbing and descending stages first decreased and then showed an increasing trend. This change could be detected by the on-body PCAM sensor. To validate this concept, a sensor-integrated robot successfully climbed over six hills of different heights (i.e., 1.5 mm, 3.0 mm, 4.5 mm), as shown in Fig. 6a, d, and Supplementary Fig. 47. Correspondingly, the recorded sensor profiles in Fig. 6d showed six distinct plateaus, with different hill heights distinguishable based on the sensor signal amplitudes (details in Methods, analyses in Supplementary Table 1).

With the collective sensing data derived from multiple robot navigations, another ANN model was trained to automatically predict the altitude of the passing terrain (detailed ANN framework is provided in Methods). Herein, 30 collected data sets were used as the training

data to develop the terrain prediction model (see data files in GitHub). Additionally, two extra test data, as shown in Fig. 6e, f (previously unseen by the prediction model) were used to evaluate the model prediction accuracy. The mean relative error (MRE) and mean absolute error (MAE) between the predicted terrain altitudes and actual altitude values were calculated in Eqs. 9 and 10, respectively.

$$MRE = \frac{1}{N}\sum_{i=1}^{N}\frac{\left|H_{pre}-H_{real}\right|}{H_{real}} \tag{9}$$

$$MAE = \frac{1}{N}\sum_{i=1}^{N}\left|H_{pre}-H_{real}\right| \tag{10}$$

where $N$ is the hill index in the terrain ($N = 1-6$); $H_{pre}$ and $H_{real}$ are the predicted and real height of $i^{th}$ hill, respectively. As evidenced in Fig. 6e, f, the MRE values on the two test sets were calculated to be 9%

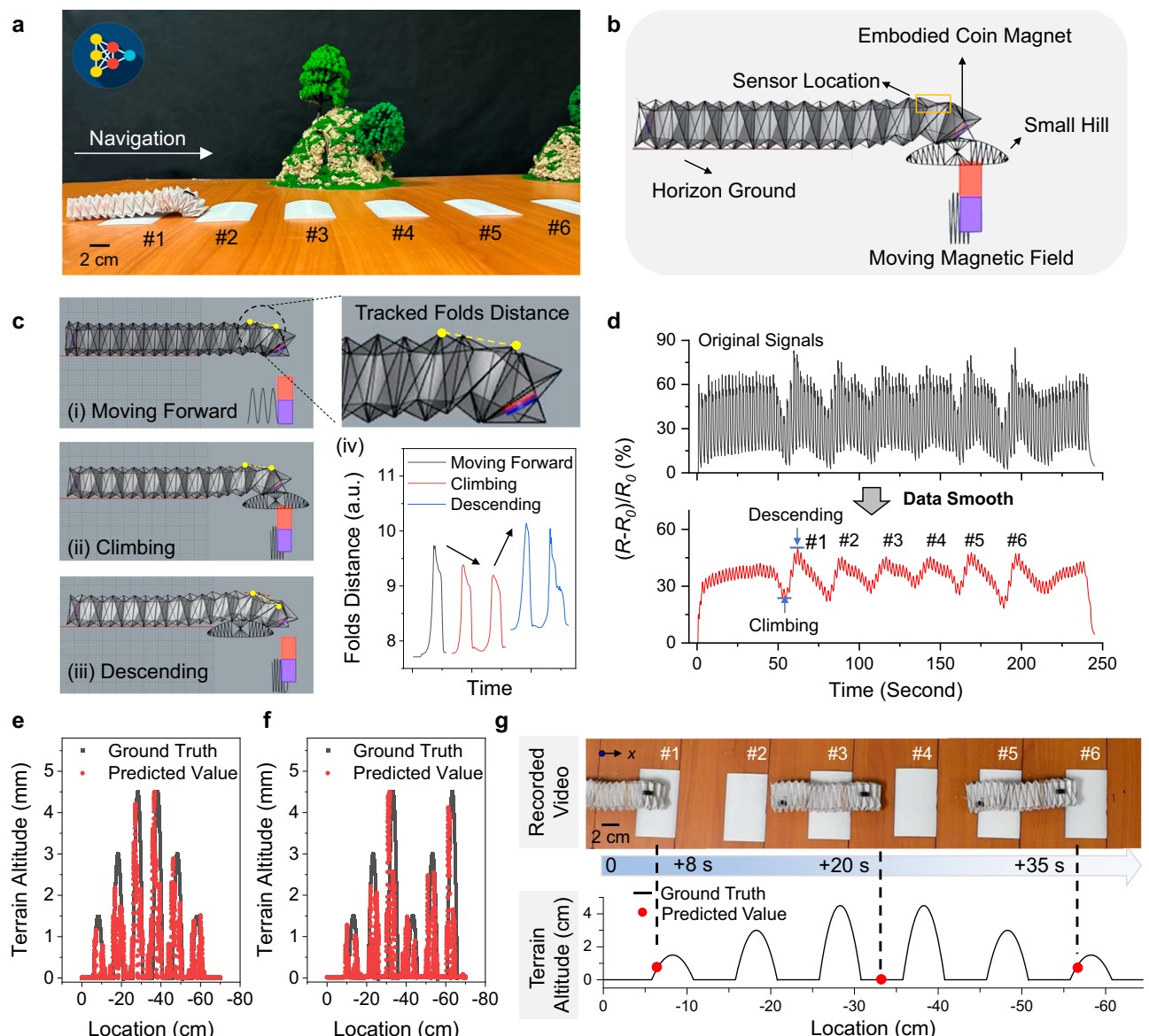

**Fig. 6 | Surrounding awareness robotic navigation. a** Digital photo of robot navigation in an artificial terrain with varying altitudes. **b** Simulation of an origami robot climbing over a small hill. **c** Simulated folds distance changes at the origami robot head during the climbing and descending stages of climbing over a hill. **d** Sensing profiles of the sensor-integrated origami robot when climbing over six different hills. Data smoothing by adjacent-averaging method, see details in Methods. **e, f** Performance of the terrain prediction model on two test sets. **g** Comparison between model-predicted terrain altitudes and the ground truth recorded by video. The time reading is the record in Supplementary Movie 10.

and 10%, respectively, with corresponding MAE values were 0.2 and 0.3 mm, when the predicted and actual altitudes were compared. As shown in Fig. 6g and Supplementary Movie 10, the terrain altitudes determined by the ANN model closely aligned to the ground truth throughout the robot exploration.

## Discussion

In this study, as shown in Fig. 7, we develop a computational strain sensor design that overcomes the stringent demands of predictive manufacturing, user-specific parameters, and ultra-stable requirements based on a programmed crack array within micro-crumples strategy. By controlling user-defined parameters, namely crack densities and shrinkage ratios, the GF and linear working windows of PCAM sensors were highly tunable, and their sensing behaviors could be modelled by FEA tools with high accuracy. The PCAM sensors exhibited excellent mechanical robustness, under various challenging operating conditions including noisy, intermittent, and dynamic

operating environments. These sensors can be further integrated into various soft robots spanning macro-micro scales, maintaining consistent and reliable perception, regardless of the robot scale. Finally, machine intelligence was realized by introducing ANN algorithm to a sensor-integrated origami robot, which demonstrated autonomous robot navigation with high-accuracy trajectory prediction, and surrounding awareness navigation.

This work addresses several longstanding challenges of strain sensors for soft robots. Firstly, with the computational design approach, strain sensing curves of target devices could be simulated by FEA models with high accuracy, which allow virtual verification without conducting resource and time-consuming experiments. The modelling approach provides a unified sensor platform that can be easily customized and extended to improve the interoperability of robotic systems across various tasks and scales. Conventionally, it is nearly impossible to predict a tactile sensor performance based on composition and morphology[15,25]. Our proposed strategy bridges the

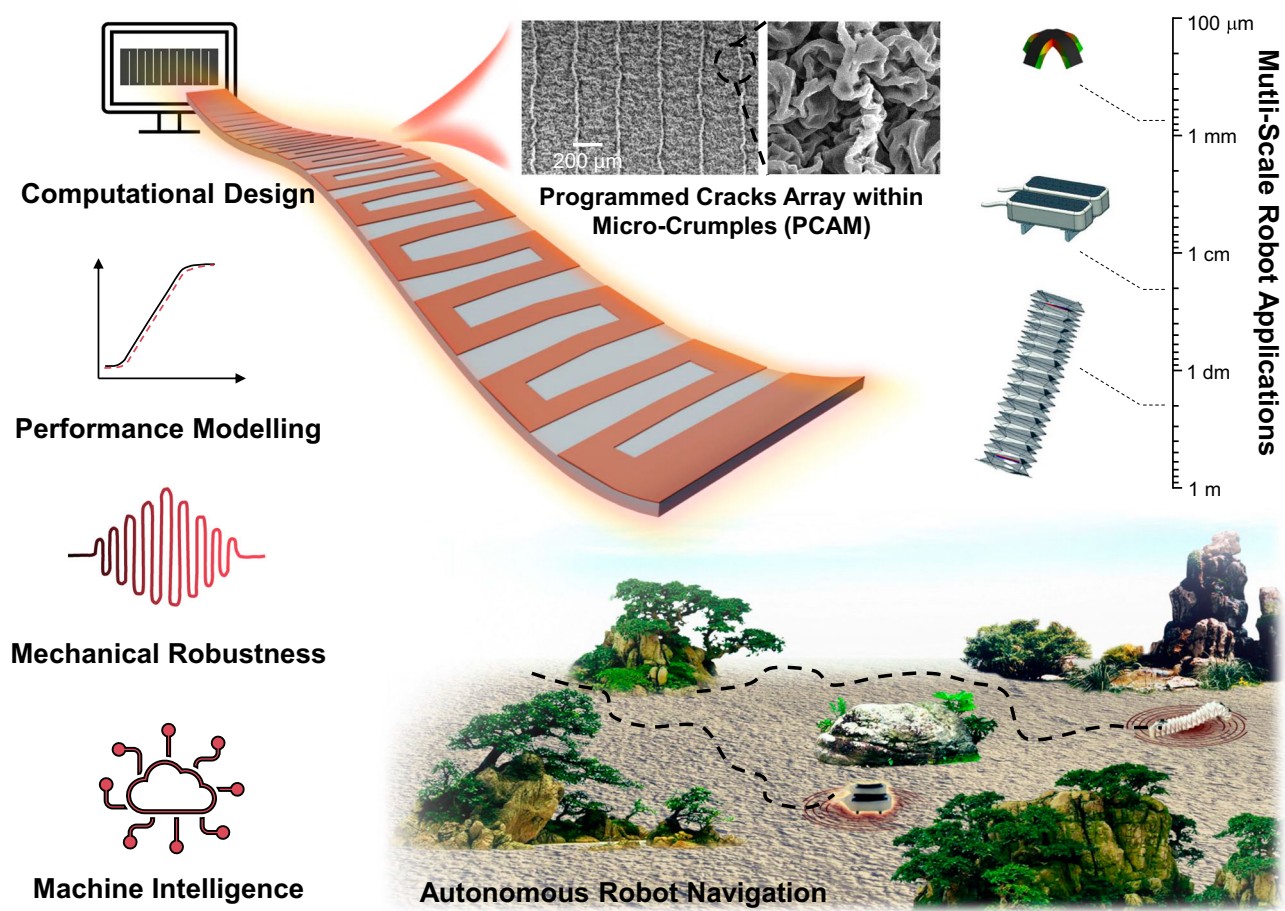

**Fig. 7 | Computational design of ultra-robust strain sensors for autonomous soft robots.** The workflow of this study includes computational sensor design, sensor performance modelling, mechanical robustness test, and machine intelligence construction. The developed PCAM sensors satisfy the sensing requirements of diverse soft robots from macro to micro scales, and machine intelligence is realized by introducing machine learning algorithm to achieves autonomous robot navigation with high accuracy.

gap between physical modelling techniques and empirical experimentation, facilitating the predictive design of new strain sensors. Compared to the state-of-the-art strain sensors[17,19−22,24,28,30,31,44,47−55], as shown in Table 1, our developed PCAM sensors showed advanced sensor characteristics and sensing curve modelling capability simultaneously. More comparisons of the pros and cons between PCAM sensors and our previous work[9] as well as recent self-healing sensors[56−62] were provided in Supplementary Notes 11 and 12.

Secondly, the achievement of ultra-robust sensor represents a significant milestone in overcoming the mechanical failure under complex and dynamic working environments of soft robots. The importance of sensor stability has been highlighted by many review papers[26,63,64], but most sensors cannot sustain the noisy, intermittent, and dynamic environments that represent the complexity and uncertainty present in real-world working situations of soft robots[22,31,34,45]. In this work, by utilizing programmed cracks array within microcrumples strategy, our developed PCAM sensors maintain robust sensing responses under various challenging operating conditions including noise interruptions (up to 50% strain), intermittent cyclic loadings (100,000 cycles), and dynamic operation frequencies (0−23 Hz). These robust sensors greatly enhance the perception capability of integrated soft robots, which provided stable sensing signals to monitor their high DOF body deformations and multi-modal actuation behaviors, ensuring high ML learning efficiency when constructing the prediction model (see detailed discussion in Supplementary Note 13).

Furthermore, the effective integration of PCAM sensors on soft robot body constructed high-level machine intelligence on complex soft crawling robots. For current ML applications on soft robots or actuators, the applied targets mainly refer to soft gloves[65−67] or soft grippers[13,68−70]. To capture their motions, there is no doubt to attach sensors on all gripper or glove fingers. However, for the crawling origami robots in this work, there are more than 40 possibilities of the sensor locations, which posed the challenge to capture its high DOF and multi-modal motions efficiently. To achieve robot autonomy, as discussed in Supplementary Note 13, a high-resolution sensor network (optimizing the sensor number and location on the robot body) was developed to collect the most representative key information of robot origami motion. As a result, simple ANN framework and less than 40 training samples were sufficient to generate the prediction model, and high-level robot autonomy on soft crawling robot (i.e., robotic trajectory prediction and topography altitude awareness) was successfully realized (see comparison of recent soft crawling robots in Supplementary Table 2).

For the future perspective, PCAM sensors show high adaptability to diverse soft robots, from origami to pneumatic, and across scales from macro to micro. These highly adaptable sensor-integrated soft machines could be applied in various environments, allowing them to have augmented perception functions as well as machine intelligence capabilities. This versatility offers advantages for a wide range of tasks, such as robots that can operate in confined physical spaces (e.g., chemical spills and cargo delivery), navigate unknown environments,

**Table 1 | Comparison of the state-of-the-art crack-based soft strain sensors**

| Crack-based strain sensor type | | Planar sensor with nanomaterials [19-21,28,30,44,47] | Sensor with crumpled or wrinkled surface [17,24,31,48-51] | Sensor with pre-stretch induced local cracks [22,49,52-55] | PCAM sensor (This work) |
|---|---|---|---|---|---|
| Typical strain-sensing profile | |  |  |  |  |
| Sensor characteristics | Linear response | ● | ● | ●●● | ●●● |
| | >100% working window | ●● | ●●● | ●● | ●●● |
| | Tuneable GF | ●●● | ●● | ●●● | ●●● |
| | Tuneable working window | ●● | ●●● | ●● | ●●● |
| Sensing curve modelling | | n/a | n/a | n/a | ●●● |
| Sensor robustness | Mechanical noises | ● | ●●● | ● | ●●● |
| | Intermittent cyclic loading | ● | ●● | ● | ●●● |
| | Dynamic working frequency | ● | ●● | ●● | ●●● |
| | High DOF deformations | ● | ●● | ●● | ●●● |
| Sensor performance reproducibility | | ● | ●● | ●● | ●●● |

Note: In this table, ●, ●●, and ●●● refer to low, medium, and high, respectively. n/a means: not available.

and enable remote control of untethered robots. Furthermore, more advanced ML algorithms could be developed to connect and manage multiple sensor-integrated robots, providing opportunities to achieve higher swarm intelligence of soft robots.

## Methods

### Materials
Single-walled carbon nanotubes (SWNT, Timesnano Co. Ltd, China), sodium dodecyl sulfate (SDS, Sigma-Aldrich, >99.9), ethanol (Thermo Fisher, >99.5%), silver nanowire (Sigma-Aldrich), and ethyl acetate (J.T. Baker; 99.9%) were used as received without further purification. Polyvinylidene fluoride (PVDF) membrane (diameter 3.8 cm, 0.22 μm pore, Merck Millipore) were purchased from Durapore. Biaxial polystyrene shrink films were purchased from Grafix. Ecoflex™ and Dragon Skin were purchased from Smooth-On, Inc. Silver paste was purchased from Ted Pella Inc. Deionized (DI) water (18.2 MΩ) was obtained from a Milli-Q water purification system (Merck Millipore) and used as the water source throughout the work.

### Preparation of SWNT dispersion
The SWNT dispersion was obtained by adding the SWNT powders into the SDS solution at the mass ratio of SWNT:SDS = 1:20. Then, the mixture was ultrasonicated for 2 h by a probe sonicator, and the concentration of the final SWNT dispersion was about 0.5 mg mL$^{-1}$.

### Preparation of freestanding SWNT layer
The as-prepared SWNT dispersion was deposited onto PVDF membranes through vacuum-assisted filtration systems. To remove SDS residues, the filtered SWNT thin films were rinsed with excessive DI water. Afterward, freestanding SWNT layers were detached from the PVDF membranes by immersing them in ethanol. The SWNT layers were stored in ethanol.

### Preparation of interdigital pattern on SWNT layer
A PS shrink film was cut into multiple rectangle-shaped pieces (5 × 5 cm²), washed with ethanol, and dried under N² flow. The cut shrink films were next treated with oxygen plasma for 10 min to enhance the hydrophilic interactions between PS substrates and SWNT layers. Afterward, the freestanding SWNT layers were carefully transferred from ethanol onto the plasma-treated shrink films followed by overnight drying. The SWNT-coated PS device was then put into the design space of a laser machine. After the setup of laser beam power (i.e., 0.08 mW), user-defined interdigital patterns that consist of the

interlocking comb-shaped crack array were sintered on SWNT-coated PS device, while the PS substrate was kept integral.

### Preparation of PCAM sensor
With the programmed interdigital crack pattern, the SWNT-coated PS device was heated in an oven at 140 °C to induce thermal contraction. By harnessing surface instability during thermal contraction, the SWNT layers were deformed with localized micro-crumples. The shrunk samples were then coated by a 2 mm thick Ecoflex (00−35). Afterward, it was immersed in an ethyl acetate bath for 24 h to dissolve the PS substrate. Next, the Ecoflex-coated SWNT device was taken out followed by air drying and wiring electrical leads to obtain the corresponding PCAM sensor. Silver paste was applied between SWNT layers and copper wires to ensure good electrical contact. The resistance profiles of PCAM strain sensors were monitored by the multimeter Keithley DMM6500.

### Preparation of planar sensor
A freestanding SWNT layer was carefully transferred onto a 2 mm thick Ecoflex (00−35) substrate in an ethanol bath followed by overnight drying. Copper wires were then connected to two ends of the planar SWNT layer, and silver paste was applied at the connection joints to ensure good electrical contact.

### Preparation of crumpled sensor
A planar SWNT layer was carefully transferred onto the plasma-treated PS shrink film followed by overnight drying. Afterward, the SWNT-coated PS device was heated in an oven at 140 °C for 6 min without constraints for biaxial shrinkage. The shrunk samples were then coated by a 2 mm thick Ecoflex (00−35). Afterward, it was immersed in an ethyl acetate bath for 24 h to dissolve the PS substrate. Next, the Ecoflex-coated SWNT device was taken out followed by air drying and wiring electrical leads to obtain the corresponding crumpled sensor.

### FEA simulation
The 3D models of PCAM sensors were built by using COMSOL Multiphysics, and their surface characteristics were modeled by the Freeform feature. Some basic simulation parameters for sensor models were set as follows: SWNT layer, conductivity 3000 S cm$^{-1}$, Poisson's ratio 0.49, mass density 1.3 g cm$^{-3}$; Ecoflex substrate, Poisson's ratio 0.49, mass density 0.9 g cm$^{-3}$. We want to note that, as listed in Supplementary Table 3, two stickiness factors should be properly chosen for the modelling tasks: (1) stickiness parameters between two crack

boundaries, and (2) stickiness parameters between SWNT and Ecoflex layers. Some methods during simulation are listed as follows: the contact is modeled based on "penalty formula"; the shear stiffness is defined using "shear to normal ratio"; the cohesive zone is modeled based on "displacement-based damage. Cartesian coordinate was chosen for the mesh method, and the minimal element size was $150 \times 150 \times 100\,\mu m$. Based on varying $\rho$ and $\varphi$ values, more modelling details of different PCAM sensors were provided in Supplementary Table 4, where the crack number and Young's Modulus of SWNT layers are the main parameters to model the sensor shape deformation. The FEA simulation was conducted as shown in Supplementary Fig. 48, where the left boundary of the 3D model was fixed, and the right boundary was set to be movable along x directions. Uniaxial stretching was simulated by moving the right boundary, and the equivalent elastic strains and the overall deformation were recorded. Meanwhile, sensor resistance was simulated and extracted by applying a constant current on the device and measuring the surface potential changes between left and right ends of the SWNT layer. The relationship between current, static electric field and electric potential is calculated by Eq. 11,

$$\begin{cases} -\nabla V = E \\ \nabla \times E = 0 \\ \nabla \cdot J = 0 \\ J = \sigma E \end{cases} \quad (11)$$

where J is the current density, E is the static electric field, $V$ is the electric potential and $\sigma$ is the conductivity. The potential difference between two boundaries is used to calculate the resistance.

### Calculation of material roughness

A roughness value can either be calculated on a profile (line) or on a surface (area). The profile roughness parameter is more common. In this work, based on profile data, we adopt arithmetical mean height ($R_a$) to quantify the material roughness. As shown in Supplementary Fig. 49, $R_a$ indicates the average of the absolute value along the sampling length, which is calculated by Eq. 12,

$$R_a = \frac{1}{l_r} \int_0^{l_r} |Z(x)| dx \quad (12)$$

where $l_r$ is the reference line length, $x$ is the location along the profile, $Z(x)$ is the height distance toward the reference line.

### Fabrication and actuation of origami robot

Based on the crease pattern in Supplementary Fig. 50, multiple origami units were folded by using cellulose paper. To enhance the mechanical stability of origami structure, these paper origamis were then surface coated by a thin layer of Ecoflex (00−35) (thickness ca. 0.5 mm). As shown in Supplementary Fig. 51, the Ecoflex-coated origami showed consistent force-strain curves under 1000 cycles of mechanical strain loading. Three Ecoflex-coated origami units were then assembled as an origami robot (see Fig. 4a), where two small Nd-Fe-B magnets (diameter, 10 mm; thickness, 1 mm) were integrated into the head and tail of robot to serve as magnet responsive units. Under a moving magnetic field, as shown in Supplementary Movie 6, the as-fabricated origami robot was able to make inchworm-like crawling motions. To realize the robot movement in a repeatable way, we established an automatic actuation system. As shown in Supplementary Fig. 42, this system consisted of a robot car, a telescopic rod, and a cubic magnet (dimension 20 mm), enabling the programmable control of a moving magnetic field in a 3D space including both horizontal and vertical directions. Supplementary Movies 6 and 9 demonstrated robot crawling capability (e.g., moving forward/backward, turning left/right, passing through small hills).

### Fabrication and actuation of pneumatic robot

The structure of the pneumatic robot was described in Supplementary Fig. 25. First, a mold for fabricating an elastomeric tube with one inner air channel was designed and created by using a 3D printer with acrylonitrile butadiene styrene (ABS). Second, a highly extensible, elastomeric material (Ecoflex(0020)) was poured into the 3D-printed mold. After 5-h curing at room temperature, the molded elastomer tube was peeled off and covered with a relatively inextensible, compliant, flat elastomeric substrate (Dragon Skin 20). At last, two elastomeric tubes were placed in parallel, which served as left and right bodies of a soft robot. Two 3D-printed friction feet were installed at both head and tail of robot body to complete the fabrication of a soft pneumatic robot. Ecoflex (0035) was used as a glue to assemble all units together. Meanwhile, in Supplementary Fig. 30, a gas pump-based pneumatic system was established for automatic robotic actuation. As shown in Supplementary Movie 7, through the selective drive of left and right robot body, the pneumatic robot was able to make inchworm-like crawling motions (e.g., moving forward, turning left/right).

### Fabrication and actuation of tetrapod microrobot

First, as shown in Fig. 4g, small-sized interdigital patterns (dimension, 2 mm) were programmed on SWNT-coated PS devices through the laser machine. Single interdigital pattern was cut out and heated in an oven at 140 °C for 6 min to induce the thermal contraction. The shrunk sample was then coated by a (Nd-Fe-B)-contained Ecoflex, which was obtained by mixing 10 wt% Nd-Fe-B particles (diameter, ~5 µm) with Ecoflex (0035). After crosslink, it was immersed in an ethyl acetate bath for 24 h to dissolve the PS substrate. Then, the (Nd-Fe-B)-Ecoflex-coated SWNT device was taken out followed by air drying to serve as a tetrapod microrobot (length/width, 700 µm). The tetrapod microrobot was put into the space between two electromagnets with high field uniformity (at 1.2 T; EM4-HVA-S, Lake Shore Cryotronics), which induced the magnetic orientations inside the tetrapods (Supplementary Fig. 39). The tetrapod microrobot was able to undergo reversible shape/body transformation under magnetic actuation and its resistance profile was monitored by the multimeter Keithley DMM6500.

### Tracking of origami robot location in the artificial terrain

In this work, we used an open-sourced software of "Tracker" to track and analyze the robotic location. First, one set of videos that recorded robotic movements in the artificial terrain was collected by a camera (see the setup in Supplementary Fig. 44). Afterward, these videos were inserted into the "Tracker" software. As shown in Supplementary Fig. 52, "Tracker" could realize automated object tracking with position and velocity, then save/output the data. Detailed tutorials about Tracker are provided at the website: https://physlets.org/tracker/".

### Modelling of origami robot

The modelling of the origami robot is performed by the software of Grasshopper. Kangaroo, a Grasshopper plugin, provides an array of physics simulation tools for Rhino 3D modelling software, including soft body physics, rigid body dynamics, and particle simulations[71]. In this study, we utilized the Kangaroo setup in Supplementary Fig. 53 to simulate the proposed origami structure's deformations. The simulation integrates structural mechanics and electromagnetics to create a multiphysics simulation that accurately models the origami structure's motions. Based on the simulated origami motion, the distance changes between designated origami folds are extracted to reflect robot body deformations during actuation.

### Dataset collection for robotic trajectory prediction model

First, training data consisted of independent and dependent variables. In this work, independent variables contained multi-channelled sensor data as well as robot actuation information, while the dependent

variable was the robot trajectory (i.e., real-time robot location in the terrain). Herein, the sensor data was recorded by the multimeter Keithley DMM6500, and robot actuation information was extracted from robot control station, while robot trajectory was recorded a camera system (see Supplementary Fig. 44), which was further extracted using a "Tracker" program. With the setup in Fig. 5b, the sensor-integrated origami robot was placed in an artificial terrain to execute different trajectories, including 7 modes: (1) moving forward $N$ steps; (2) moving forward $N$ steps then meeting an obstacle with $N$ steps; (3) moving forward $N$ steps then turning left $N$ steps (meeting an obstacle with $N$ steps or not); (4) moving forward $N$ steps then turning right $N$ steps (meeting an obstacle with $N$ steps or not); (5) moving forward $N$ steps then moving backward $N$ steps; (6) moving forward $N$ steps then moving backward $N$ steps then turning left $N$ steps (meeting an obstacle with $N$ steps or not); (7) moving forward $N$ steps then moving backward $N$ steps then turning right $N$ steps (meeting an obstacle with $N$ steps or not). By randomly setting different $N$ values, we collected 43 datasets which were not repeated. As shown in Fig. 5d, we selected 5 datasets with representative trajectories as testing datasets and took other 38 datasets as training datasets.

### Dataset collection for terrain prediction model
First, training data consisted of independent and dependent variables. In this work, independent variable was on-body sensor data, and the dependent variables were the robotic trajectory (i.e., real-time robot location) as well as the terrain information (i.e., terrain altitude changes along the robotic trajectory). Herein, the sensor data was recorded by the multimeter Keithley DMM6500, and robot trajectory was recorded a camera system (see Supplementary Fig. 44) which was further extracted using a "Tracker" program, and the terrain consisted of 6 different small hills (2 hills with 1.5 mm height, 2 hills with 3.0 mm height, 2 hills with 4.5 mm height) and their distribution information was recorded by the researchers. With the setup in Fig. 5a, a sensor-integrated origami robot was placed in the artificial terrain to climb over 6 hills with different combinations, such as 1.5-3.0-1.5–4.5-3.0-4.5, 4.5-3.0-1.5-4.5-3.0-1.5, and so on. By randomly setting different hill combinations, we collected 32 datasets which were not repeated. As shown in Fig. 6e, we selected 2 datasets with representative hill distributions as testing datasets and took other 30 datasets as training datasets.

### ANN model for robotic trajectory prediction
As shown in Supplementary Fig. 54, the ANN model is constructed by a bunch of neuron base units with learning parameters (e.g., $w_{12}$, $w_{15}$). Similar to other supervised ML algorithms, ANN model aims to use a set of training data to predict an output based on the inputs. In this study, the inputs are the multi-channeled PCAM sensor data as well as the actuation information, and the outputs are the predicted $x$ and $y$ locations in the terrain in time series. The entire network consisted of 4 fully-connected layers, which were serialized by adding an activation function, Relu, and a Batch Normalization layer, in between each adjacent layer. The mean squared errors between the predicted values and the ground truth values were utilized as the model's loss function. A 10-fold cross-validation training strategy was adopted during the training of the ANN model. The optimization process of k-fold value in this ML task was discussed in Supplementary Note 14. Afterward, a set of test data was used to evaluate the prediction accuracy of the trained ANN model. The source code of the ANN model was implemented in Python using the Keras and TensorFlow frameworks, which is publicly available on GitHub (https://github.com/SS47816/strain_sensor_prediction).

### ANN model for terrain altitude prediction
For this task, the inputs are the PCAM sensor data, and the outputs are the predicted terrain altitudes in time series. Similar to the network

model used in the previous trajectory prediction task, a network consisting of 4 fully-connected layers, with Relu activation function and Batch Normalization layer in between was trained and tested for the altitude prediction. The mean squared errors between the predicted values and the ground truth values were utilized as the model's loss function. A 10-fold cross-validation training strategy was also adopted during the training of the ANN model. Afterward, a set of test data was used to evaluate the prediction accuracy of the trained ANN model. The source code of the ANN model was implemented in Python using the Keras and TensorFlow frameworks, which is publicly available on GitHub (https://github.com/SS47816/strain_sensor_prediction).

### Data smoothing by adjacent-averaging method
The smoothing method used in this study is adjacent-averaging, where each smoothed data point is computed from the data points within a moving window. For example, let $\{f_i,|,i=1,2,\ldots,N\}$ be the input data points and let $\{g_i,|,i=1,2,\ldots,N\}$ denote the output data points. Then, each $g_i$ is computed by Eq. 13,

$$g_i = \frac{1}{m}\sum_{i-m/2}^{i+m/2} f_i \tag{13}$$

where $m$ is the value of the Points of Window variable ($m = 100$ in this work).

### Characterization and measurements
The surface morphologies of SWNT layers were characterized by using a SEM (FESEM, JEOL FEG JSM 7001 F). Atomic force microscope (AFM) images of SWNT layers were collected on a commercial scanning probe microscope (SPM) instrument (MFP-3D, Asylum Research, CA, USA). Optical microscope images were captured by an Olympus BX53M microscope. The characteristic dimension changes of PS shrink films were quantified by using *ImageJ*. The tensile strain-stress tests were implemented using the tensile machine (MultiTest 1-i) with a 5-N load cell. The multi-channeled resistance profiles of robotic sensors were monitored by the multimeter Keithley DMM6500.

### Data availability
The data generated in this study are provided in the Supporting Information/Source Data file. Sensor data files for model training and testing generated in this study have been deposited in the public GitHub (https://github.com/Haitao008/Supporting-Materials) without any restrictions. The data that support the plots within this paper and other findings of this study are available from the corresponding authors upon request. Source data are provided with this paper.

### Code availability
The Python code to implement the machine learning tasks in this study has been deposited in the public GitHub (https://github.com/SS47816/strain_sensor_prediction) and Zenodo (https://doi.org/10.5281/zenodo.10464057)[72] without any restrictions.

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

## Acknowledgements

The authors express their gratitude to Dr. Wei Li Ong and Dr. Wanheng Lu for assistance during characterization and visualization. The authors acknowledge support from the A*STAR RIE2025 Manufacturing, Trade and Connectivity (MTC), M22K2c0081 (to G.W.H.), Fundamental Research Funds for the Central Universities (23SH0201313 and W099001 to H.Y.), Advanced Research and Technology Innovation Centre (ARTIC), the National University of Singapore under Grant (project number: A-0005947-24-00 to G.W.H.).

## Author contributions

H.Y. and G.W.H. conceived the concept and planned the project. H.Y. conducted the sensor experiments and data analyses. S.D. and H.Y. designed the soft robot structures and robot actuation system and conducted robot dynamics characterization. J.W. and H.Y. developed the FEA simulation methods for all sensors. S.S. helped with the ML model optimization, R.S. performed the origami simulation. S.W.L.N. and X.P. programmed the laser and mechanical tests. H.Y. and G.W.H. wrote the manuscript. All authors have given approval to the final version of the manuscript.

## Competing interests

The authors declare no competing interests.
