## [Peer Review File · Nature Communications]

REVIEWER COMMENTS

Reviewer #1 (Remarks to the Author):

The paper presented a computational sensor design method based on programmed crack array that enables the production of strain sensors. The paper is interesting and also include machine learning method for the design and trajectory planning. The results look promising and could be used for a variety of applications. However, concerns are listed as below.

1. The novelty of the work is not very clear. There are plenty of work in developing strain (or multifunctional) sensors in the field. The advantages and novelty of the proposed sensor were not clear.
2. The authors claimed that "well-controlled sensor dynamics" while there is no evidence on this. Only some static calculations are presented in the paper. Dynamic sensing is quite a research challenge in robotic engineering. However, it is very confusing to use "sensor dynamics" as "dynamics" is usually more about actuators (actuation) and motion.
3. Machine learning looks quite interesting here and has contributed to the design of the sensor. However, this is more of an application/programme based contribution, and the novelty here is not particularly significant. Researchers have been developed soft robots or actuators based on machine learning technology too. What's the difference of the methodologies between the proposed and existing?
4. Some more technical questions, Figure 2, physical modelling of the PCAM sensor. Figure 2f is not clear, it would be good to explain the red (55%) and (25%) actually present different trend/slope when the strain increasing. Any reason cause this?
5. The motivation of developing the origami robot is not clear. Is this for the demonstration only? Why origami robot?

The work is interesting, but need some improvements before publishing as a journal paper.

Reviewer #2 (Remarks to the Author):

The authors present a strain sensor with computational design to enable predictable and robust sensing performance. Overall the authors carried out thorough and in-depth discussion and computation-assisted optimization of the sensor. Nevertheless, I found the materials design concept and sensing mechanism are similar to Ref. [9] from the same institution, with rather incremental enhancement.

The authors claimed the demonstration of a computational strain sensor design. But in fact, any good resistive strain sensor can be implemented in the algorithm. Despite beautiful demonstration, from materials point of view, the novelty is not very high.

Very recently there have been a number of reports showing soft materials with self-healing actuation (e.g., <https://www.nature.com/articles/s41563-020-0736-2>), which is clearly more advanced than the system presented here. Eventually with cracks propagation after extensive cycles, the self-healing materials would outperform.

Overall I consider the novelty of this work does not represent a milestone in this field and is not sufficient to warrant the publication in Nat. Comm.

Reviewer #3 (Remarks to the Author):

In this study, authors developed a sensor using the SWCNT-ecoflex composite by introducing Programmed cracks with an interlocking comb-shaped pattern. This sensor underwent heat treatment to achieve the desired shrinkage (i.e. pre-strain). The findings revealed that its performance, including sensitivity and linear working window, can be modulated using the crack density (ρ) and shrinkage ratio (ϕ). Both experimental results and Finite Element Method (FEM) analysis supported these findings. The team also put the sensor to the test under various conditions, such as dynamic mechanical loading, cyclic loading, and operation frequency tests, to ascertain its stability. All results indicated a robust stability. Furthermore, the versatility and robustness of this sensor were highlighted when it was successfully integrated into multiple soft robot applications. The team demonstrated its compatibility across a range of robotic scales: from large-scale origami robots to pneumatic robots and even tiny microrobots. In an intriguing application, the sensor, once attached to an origami robot, provided data that helped train an Artificial Neural Network (ANN) to predict the robot's movement and also to gauge the height of potential obstacles. The paper presents a set of multiple intriguing results, yet there's room for enhancement by considering the subsequent feedback.

- 1) Figure 1e demonstrates the excellent reproducibility of the sensor. However, it only showcases scenarios with low crack density ($\rho=300\mu\text{m}/\text{mm}$). One might anticipate that as the crack density increases, the performance uncertainty may also rise. It would be beneficial to validate the reproducibility for higher crack densities. Alternatively, adding error bars to the results in Figure 1g for instances of higher crack density and discussing them would be advantageous.
- 2) In the Sensor design section on p.6, it's mentioned that sensitivity can be modulated by the crack density, and Figure 1g suggests that the linear working range remains fairly consistent across varying crack densities. However, it might appear that the linear working range expands as crack density increases. To prevent this potential misunderstanding, quantifying the linear working range might be a prudent approach.
- 3) Sensor performance can be assessed based on (i) sensitivity, (ii) linear working range, (iii) resilience under cyclic loading, and (iv) hysteresis during loading and unloading phases. While the authors have provided encouraging results for (i) through (iii), there's an absence of information regarding hysteresis. Incorporating hysteresis data would enhance the evaluation.
- 4) Pertaining to point 3), the simulation was solely conducted for tensile loading and omitted the unloading phase. It would be valuable if the authors could incorporate a simulation illustrating the sensor's response during unloading. If this poses a challenge, a brief commentary on its practicability would be appropriate.
- 5) In the 'Intelligent sensor network for robotic trajectory prediction' section, the authors reference the use of actuation information and sensing information for ANN training. Delving into the specifics of what comprises the actuation information would be insightful. Moreover, both this section and the 'Surrounding awareness robot navigation' section indicate the use of 38 and 30 training data points respectively. An explanation of the methodology behind generating the initial training data would enhance understanding. Also, the decision to use a 10-fold validation on a dataset comprising roughly 30 samples may seem disproportionate, leading one to question if the selected k-value is overly high for the dataset size.
- 6) Considering the fabrication procedure, it appears that the electrically percolating SWCNT layer might not be completely encased within the ecoflex matrices. A sandwiched structure could potentially decrease hysteresis and improve wear resistance. Including a discussion on this perspective would be valuable.
- 7) Concerning the underlying mechanism of the 'delta R-strain' response, the authors suggest a gradual appearance of the surface crack with the increase of strain. However, I anticipate that the entire surface crack path would open abruptly at the applied strain where the compressive pre-strain from initial thermal contraction is nullified. Thus, from zero strain to this critical strain point, the resistance would see a gradual rise (as seen from 0-20% strain in Fig 1e). Once the surface crack emerges, a stiffer resistance increase would ensue due to the elongated serpentine-like electrical path (as observed from 20-60% strain in Fig 1e). Introducing a stress-strain curve (from experiments and simulation) in the discussion might offer more clarity in testing this perspective.

Responses to the Reviewers' Comments

Reviewer #1

Comment 1: *The paper presented a computational sensor design method based on programmed crack array that enables the production of strain sensors. The paper is interesting and also include machine learning method for the design and trajectory planning. The results look promising and could be used for a variety of applications. However, concerns are listed as below.*

Response 1: We thank Reviewer #1 for his/her careful review and positive evaluation on our work. We have made point-by-point responses for each comment below.

Comment 2: *The novelty of the work is not very clear. There are plenty of work in developing strain (or multifunctional) sensors in the field. The advantages and novelty of the proposed sensor were not clear.*

Response 2: To respond Reviewer #1's concerns, we are keen to highlight the advantages and novelty of the developed PCAM sensor are at three aspects, including **(1) accurate sensor performance modelling, (2) excellent sensor signal robustness, and (3) high-level soft robot integration.**

Our first contribution is the implementation of well-programmed crack propagation and finite element analysis (FEA) to build up a high-accuracy modelling tool for strain sensors, capable of predicting sensor's sensing curves without conducting trial-and-error experiments. As mentioned in some important articles (*Sci. Robot.*, 2019, 4, eaav1488; *Nat. Mach. Intell.*, 2022, 4, 194–195.), one critical challenge is to predict a soft strain sensor performance based on composition and morphology, which is hindered by the unpredictable material and structure evolutions under deformations of conventional sensors. In this work, we tackled this challenge by the laser-programmed cracks array within micro-crumpled strategy. The resulting sensor showed accurate structure manufacture as well as well-controlled crack propagation behaviors under strains, which could be tracked and simulated by FEA model and output the sensing curves with high accuracy. This strategy bridges the gap between physical modelling techniques and empirical experimentation, facilitating the predictive design of new strain sensor with various characteristics, which has not been reported in the literature to the best of our knowledge.

Our second innovation is the development of a highly robust soft strain sensor that could work under complex operation environments. The importance of sensor stability has been highlighted by many review papers (*Adv. Mater.*, 2021, 33, 2004782; *Adv. Mater.*, 2019, 31, 1805921; *ACS Nano*, 2023, 17, 5211–5295.), but most sensors cannot sustain the noisy, intermittent, and dynamic environments that represent the

complexity and uncertainty present in real-world working situations (*Nat. Electron.*, 2022, 5, 784–793; *Nat. Commun.*, 2022, 13, 5311; *Adv. Mater.*, 2022, 34, 2203650; *Adv. Mater.*, 2019, 31, 1903789.). In this work, by utilizing programmed cracks array within micro-crumpled strategy, our developed PCAM sensors maintain robust sensing responses under various challenging operating conditions including noise interruptions (up to 50% strain), intermittent cyclic loadings (100,000 cycles), and dynamic operation frequencies (0-23 Hz).

Our third advancement is the effective integration of these sensors on soft crawling robots to enhance robot autonomy. Motion estimation of soft robot is a longstanding challenge due to its typical high-degree-of-freedom soft bodies and multi-modal locomotion (*Sci. Robot.*, 2019, 4, eaav1488, *Adv. Sci.*, 5, 2018, 1800541.), and most soft strain sensors experience material/structure failures when monitoring these dynamic robot motions. In this work, our developed PCAM sensors show robust sensor signals that satisfy the sensing requirements of diverse soft robots from origami to pneumatic, and across scales from macro to micro. After tight integration with soft robots, these robust sensor signals are helpful to achieve autonomous robot navigation with the assistance of machine learning (e.g., trajectory and terrain prediction) which has been suggested by multiple important review articles (*Sci. Robot.*, 2020, 5, eaaz9239; *ACS Nano*, 2023, 17, 5211–5295; *Adv. Funct. Mater.*, 2021, 31, 2008807.), and our work successfully realized this approach for the first time.

To better illustrate the advantages and novelty of the developed PCAM sensor, we also added Table 1 and Supplementary Table 2 to compare the sensor characteristics and robot perception/autonomy level by different kind of sensors, respectively.

Revision made

Page 15 and 16 of the revised manuscript.

Table 1 Comparison of the state-of-the-art crack-based soft strain sensors.

Crack-based strain sensor type		Planar sensor with nanomaterials	Sensor with crumpled or wrinkled surface	Sensor with pre-stretch induced local cracks	PCAM sensor (This work)
Typical strain-sensing profile					Sensor characteristics	Linear response	●	●	●●●	●●●
	>100% working window	●●	●●●	●●	●●●
	Tuneable GF	●●●	●●	●●●	●●●
	Tuneable working window	●●	●●●	●●	●●●
Sensing curve modelling		n/a	n/a	n/a	●●●
Sensor robustness	Mechanical noises	●	●●●	●	●●●
	Intermittent cyclic loading	●	●●	●	●●●
	Dynamic working frequency	●	●●	●●	●●●
	High DOF deformations	●	●●	●●	●●●
Sensor performance reproducibility		●	●●	●●	●●●

Note: In this table, ●, ●●, and ●●● refer to low, medium, and high, respectively. n/a means: not available.

Supplementary Table 2. The state-of-the-art soft crawling robot perception by mechanical sensors.

Reference	Robot sensor type	Multimodal locomotion	Basic soft robot perception			Advanced soft robot perception		Across-scale soft robot adaptability
			On-body sensing	Obstacle detection	Surface roughness classification	Trajectory prediction	Surrounding awareness	
H. Cui, et al. Science , 2022, 376, 1287.	Piezoceramic sensor	√	√	√	×	×	×	×
Y. Zhao et al., Sci. Robot. , 2021, 6, eabd5483.	Piezoresistive sensor	×	√	√	×	×	×	×
H. Yang, et al. Sci. Robot. , 2020, 4, eaax7020.	Piezoresistive sensor	√	√	×	×	×	×	×
H. Bai, et al. Sci. Adv. , 2022, 8, eabq2104.	Waveguide sensor	√	√	×	×	×	×	×
X. Wang, et al. Adv. Mater. , 2020, 32, 2000351.	Piezoresistive sensor	√	√	×	×	×	×	×
K. Li, et al. Adv. Funct. Mater. , 2022, 32, 2110534.	Piezoresistive sensor	√	√	×	×	×	×	×
H. Dong, et al. Soft Robot. , 2022, 6, 1198.	Piezoresistive sensor	√	√	√	×	×	×	×
Y.-H. Lin, et al. Adv. Intell. Syst. , 2021, 2000244.	Piezoresistive sensor	√	√	×	×	×	×	×
R. Goldoni, et al. ACS Appl. Mater. Interfaces , 2020, 12, 43388.	Piezoresistive sensor	√	√	√	×	×	×	×
This work	Piezoresistive sensor	√	√	√	√	√	√	√

Comment 3: *The authors claimed that "well-controlled sensor dynamics" while there is no evidence on this. Only some static calculations are presented in the paper. Dynamic sensing is quite a research challenge in robotic engineering. However, it is very confusing to use "sensor dynamics" as "dynamics" is usually more about actuators (actuation) and motion.*

Response 3: We thank Reviewer #1's valuable comment. To avoid potential misleading, we replaced "sensor dynamics" by "sensor structure changes under strains" in the revised manuscript.

Revision made

Page 2 of the revised manuscript.

Comment 4: *Machine learning looks quite interesting here and has contributed to the design of the sensor. However, this is more of an application/programme based contribution, and the novelty here is not particularly significant. Researchers have been developed soft robots or actuators based on machine learning technology too. What's the difference of the methodologies between the proposed and existing?*

Response 4: Thanks Reviewer #1's insightful comment. We agree that machine learning (ML) here is a general mathematical tool to analyze the sensing data and construct the prediction model. Although we did not develop a novel ML framework, our contribution here is the **enhancement of learning efficiency from the sensor aspects, including (1) robust sensor signals and (2) high-resolution robotic sensor network.**

ML allows for fast data analysis and prediction, yet it usually requires a large dataset for model training. Previously, researchers have developed smart gloves or textiles that coupled strain sensors and ML algorithm to identify different hand gestures or body poses (*Nat. Electron.*, 2021, 4, 193–201; *Nat. Electron.*, 2020, 3, 563–570; *Nat. Electron.*, 2020, 3, 571–578; *Nat. Commun.*, 2022 13, 5311.). However, as shown in Supplementary Table 8, these systems required a large amount of training data over than 300 samples. In this work, our developed PCAM sensor showed robust sensing signals with low signal deviation, which greatly improved learning efficiency of artificial neural network. The trajectory prediction model was constructed with 96% accuracy by using only 38 training samples.

We understand that it may not be entirely fair comparison between the references in Supplementary Table 8, due to their different ML tasks and application scenarios. To better investigate the effects from sensor data quality, we conducted additional ML experiments. As shown in Supplementary Figs. 33, 35, and 37, three types of sensors including PCAM sensor, crumpled sensor, and planar sensor were integrated into soft

robot to perform a 25-cm turning trajectory for five times. While the signal deviations of PCAM sensor remained small (ca. 6%), those for the crumpled and planar sensors gradually increased to >50%. By using four iteration data as training set and adopting one iteration data as the testing set to construct the trajectory prediction model (see training and testing data files in GitHub). The model performance is evaluated by mean square error (MSE), where a lower MSE value indicates better model performance. As shown in Supplementary Figs. 64 and 65, the PCAM sensor-based prediction mode showed three times lower MSE and more accurate trajectory prediction on the test set, illustrating the advantage of high learning efficiency from robust sensor data.

On the other hand, high-resolution sensor network provides another approach to improve the learning efficiency, which were realized by minimizing the number of sensors on the key locations of robot body. For current ML applications on soft robots or actuators, the applied targets mainly refer to soft gloves (*Nat. Electron.*, 2021, 4, 193–201; *Nat. Electron.*, 2020, 3, 563–570; *Nat. Electron.*, 2020, 3, 571–578.) or soft grippers (*Sci. Robot.*, 2019, 4, eaax2198; *Sci. Robot.*, 2016, 1, eaai7529; *Proc. Natl. Acad. Sci. U.S.A.*, 2020, 117, 25352–25359; *Nat. Commun.*, 2020, 11, 5381.). To capture their motions, there is no doubt to attach sensors on all gripper or glove fingers. However, for the crawling origami robots in this work, there are >40 possibilities of the sensor locations. Therefore, we need to optimize the sensor number and location on the robot body to collect the most representative key information of robot motion. For example, in the added Supplementary Fig. 66, sensor at position 1 showed higher sensitivity than position 2 when tracking the robot motions. The simulation results in Supplementary Fig. 61 indicated that applying two sensors at origami head or tail is efficient for distinguish the turning direction. Therefore, to identify moving forward/backward and turning left/right motions, the minimized number and location of the sensors on robot was optimized as Supplementary Fig. 41. Besides, for the terrain prediction task, based on simulation results in Fig. 6a-6d, instead of four sensors at both robot head and tail, one sensor at head position is enough to distinguish the terrain height changes. These efforts simplified the relationship between sensor signals and robot motion to improve the learning efficiency. Therefore, simple ANN framework and <40 training samples were sufficient to generate the prediction model. To the best of our knowledge, as summarized in Supplementary Table 2, it is the first report to achieve robot autonomy on soft crawling robot (i.e., robotic trajectory prediction and topography altitude awareness).

We have added the above discussion in the revised manuscript and expect the above approaches provides new alternatives to enhance learning efficiency of ML and construct machine intelligence on more complex soft robot system.

Revision made

Page 15 and 16 of the revised manuscript and Page 70, 71, 72, 75, 83, and 84 of Supporting Information.

Supplementary Table 8. Comparison between this work and other recent works that coupled strain sensors and ML algorithm to identify different hand gestures or body pose.

Reference	Task of ML model (prediction accuracy)	Number of model training sample
Y. Luo, et al. Nat. Electron. 2021, 4, 193–201.	Human pose classification (99.7%)	72,336
M. Wang, et al. Nat. Electron. 2020, 3, 563–570.	Hand gesture classification (100%)	3,000
Z. Zhou, et al. Nat. Electron. 2020, 3, 571–578.	Hand gesture classification (98.6%)	660
H. Yang, et al. Nat. Commun. 2022 13, 5311.	Human pose classification (100%)	300
This work	Soft robot trajectory prediction (96%)	38

Supplementary Fig. 35. Sensing performance of a PCAM sensor-integrated

pneumatic soft robot under trajectory 1, which was repeated 5 times. (a) Digital photo of robot trajectory 1. **(b)** Comparison of signal deviations of pneumatic soft robots, which integrated PCAM sensor, planar sensor, and crumpled sensor, respectively. Herein, the signal deviations were calculated based on the sensing profiles between 1st and 5th iterations under trajectory 1. Detailed calculation method was provided in Equation 6 in main manuscript.

Supplementary Fig. 33. Sensing profiles of a PCAM sensor-integrated pneumatic

soft robot under trajectory 1, which was repeated 5 times. Red curves are sensor signals from right robot body and the black curves are sensor signals from left robot body.

Supplementary Fig. 37. Sensing profiles of a planar and crumpled sensor-integrated pneumatic soft robot under trajectory 1, which was repeated 5 times. (a) Sensing profile of a crumpled sensor-integrated pneumatic soft robot in the 1st iteration of trajectory 1. **(b)** Sensing profile of a crumpled sensor-integrated pneumatic soft robot in the 5th iteration of trajectory 1. **(c)** Sensing profile of a planar sensor-integrated pneumatic soft robot in the 1st iteration of trajectory 1. **(d)** Sensing profile of a planar sensor-integrated pneumatic soft robot in the 5th iteration of trajectory 1. Red curves are sensor signals from right robot body and the black curves are sensor signals from left robot body.

Supplementary Fig. 64. MSE of trajectory prediction model based on the training data from PCAM sensor, crumpled sensor, and planar sensor.

Supplementary Fig. 65. Prediction results of trajectory prediction model trained by data based on PCAM sensor, crumpled sensor, and planar sensor, respectively.

Supplementary Fig. 66. Sensor position optimization on origami robot. When monitoring the robot crawling motion, PCAM sensor at position 1 showed much higher signal response than position 2 thus is selected as the target position for constructing the body sensor network.

Supplementary Fig. 61. Simulation of origami deformation during turning. (a) Simulated movements of origami robot during moving forward and turning left. (b) Extracted folds distance changes between the designated two folds at both left and right sensor locations during moving forward and turning left.

Supplementary Fig. 41. Origami robot with four on-body sensors.

Fig. 6 Surrounding awareness robotic navigation. (a) Digital photo of robot navigation in an artificial terrain with varying altitudes. (b) Simulation of an origami robot climbing over a small hill. (c) Simulated folds distance changes at the origami robot head during the climbing and descending stages of climbing over a hill. (d) Sensing profiles of the sensor-integrated origami robot when climbing over six different hills. Data smoothing by adjacent-averaging method, see details in Methods. (e) (f)

Performance of the terrain prediction model on two test sets. (g) Comparison between model-predicted terrain altitudes and the ground truth recorded by video. The time reading is the record in Supplementary Video 10.

Comment 5: *Some more technical questions, Figure 2, physical modelling of the PCAM sensor. Figure 2f is not clear, it would be good to explain the red (55%) and (25%) actually present different trend/slope when the strain increasing. Any reason cause this?*

Response 5: We thank Reviewer #1's insightful comment. In the revised manuscript, we added more discussions to analyze the results in Fig. 2f.

“Firstly, in Fig. 2f, the stress-strain curves of the SWNT layers with different ϕ values were measured by a tensile test machine, and the curve slope represents the Young's modulus of SWNT layers. According to the results, when ϕ increases to 25%, 40%, 55%, the Young's modulus of SWNT layer showed a decreasing trend from 33.8 to 5.5 MPa. Lower Young's modulus means the deformation of SWNT layer require smaller force. Such a trend is attributed to the different sizes of micro-crumple within the SWNT layer. As shown in Supplementary Fig. 9, micro-crumples were generated after the dimension shrinkage during the sensor fabrication. This kind of micro-structures stored elastic force within the SWNT layer. With higher ϕ values, the size of micro-crumples was enlarged, corresponding to a higher stored elastic force like a more compressed spring. Therefore, SWNT layer with higher ϕ value requires a smaller force to induce the deformation, which is reasonable to illustrate a lower Young's modulus.”

Revision made

Page 8 of the revised manuscript and Page 15 of Supporting Information.

Supplementary Fig. 9. Micro-crumple structure of PCAM sensor with different ϕ values. With higher ϕ values, the size of micro-crumples was enlarged, which was corresponding to a higher stored elastic force like a more compressed spring.

Comment 6: *The motivation of developing the origami robot is not clear. Is this for the demonstration only? Why origami robot?*

Response 6: In this work, we fabricated three kinds of soft robots, including origami robot, pneumatic robot, and microrobot, and the origami robot is selected in the demonstration of robot autonomy (i.e., trajectory prediction and topography altitude awareness) mainly due to two reasons including **(1) mature robot modelling tool** and **(2) advantageous actuation behaviors**.

Firstly, the Grasshopper software, a user-friendly modelling tool enables simulation of the body deformations of origami robot during movements (see details in Methods section). This capability allows for accurate interpretation of the integrated robot sensor signals, which is helpful for users to prepare the initial training datasets for developing the model of robot autonomy. For example, when constructing the terrain prediction model, based on the simulation results in Fig. 6c, we understand the signals differences in Fig. 6d indeed reflected the origami body deformations when climbing over six different hills instead of signal fluctuations, which could serve as efficient training data to construct the prediction model. However, similar robot modelling tool for pneumatic robots or microrobots are not available.

Secondly, origami robot showed advantages regarding multi-modal locomotion and untethered actuation, which are key aspects for robot autonomy with higher motion freedom. As summarized in Supplementary Table 5, compared to pneumatic robots and microrobots, the origami robot has large number of locomotion modes as well as untethered actuation features, simultaneously.

Considering the above factors, the origami robot was selected in the robot autonomy application/demonstration. We have added these discussions in the revised manuscript.

Revision made

Page 12 of the revised manuscript and Page 74, 80, and 81 of Supporting Information.

Supplementary Table 5. Comparison among the features of origami robot, pneumatic robot, and microrobot in this work.

	Origami Robot	Pneumatic Robot	Microrobot
Modelling Tool	Grasshopper software	Not available	Not available
Locomotion Mode	1) Moving forward 2) Moving backward 3) Turning left 4) Turning right 5) Climbing over hills	1) Moving forward 2) Turning left 3) Turning right	1) Moving forward 2) Moving backward 3) Turning left 4) Turning right
Actuation Mode	Untethered actuation	Tethered actuation	Untethered actuation

Comment 7: *The work is interesting, but need some improvements before publishing as a journal paper.*

Response 7: Thanks Reviewer #1's careful review and valuable comments. We revised and improved the manuscript by adding extra supplementary experiments and discussions, including (1) the advantages and novelty of the proposed sensor; (2) the difference of the ML methodologies between the proposed and existing; (3) the reason of selecting origami robot as demonstration; (4) results explanation of Fig. 2f; and (5) item replacement of "sensor dynamics". Overall, we believe that the story is much clear, and novelty is stronger. We hope that the revision is satisfactory to Reviewer #1.

Reviewer #2

Comment 1: *The authors present a strain sensor with computational design to enable predictable and robust sensing performance. Overall the authors carried out thorough and in-depth discussion and computation-assisted optimization of the sensor.*

Response 1: We thank Reviewer #2 for his/her positive evaluation on our work.

Comment 2: *Nevertheless, I found the materials design concept and sensing mechanism are similar to Ref. [9] from the same institution, with rather incremental enhancement.*

Response 2: We agree with Reviewer #2 that the sensing mechanism of our work and ref. [9] (*Sci. Robot.*, 2019, 4, eaax7020.) are both based on crack propagation, but their materials design concepts are totally different and the sensing characteristics of the PCAM sensor was improved a lot.

For ref. [9], the sensing materials were porous noble metals templated from ion-loaded cellulose paper, which realized effective multifunction integration of fire retardancy, resistive heating, sensing, and communication. However, from the perspective of sensing performance, it suffered from many disadvantages, including low sensitivity, fixed working window, poor sensing stability, and an inability to model its sensing performance. These issues were attributed to the uncontrolled crack growth and propagation of the templated materials under deformations, which is also a long-standing challenge for crack-based strain sensors (*Sci. Robot.*, 2019, 4, eaav1488; *Nat. Mach. Intell.*, 2022, 4, 194–195.). In this work, by introducing the programmed cracks array within micro-crumple strategy, the crack array is pre-defined by a laser machine and the crack propagation routes were well regulated. As a result, compared with ref. [9] in Supplementary Table 6, PCAM sensor showed 150 times higher gauge factor (GF) and tunable working window from 5%-50% to 20%-120%. Meanwhile, the deterministic crack propagation enabled accurate sensor performance modelling by using the FEA tool. These sensor characteristics are important for soft robot applications:

(1) High sensor sensitivity enables the robot to distinguish small body deformations during navigation tasks like climbing over a small hill in this work.

(2) Tunable sensor working window could satisfy the sensing requirements of diverse soft robots from origami to pneumatic, and across various scales.

(3) The FEA modelling tool allows virtual sensor performance verification without conducting trial-and-error experiments, facilitating the custom production of sensor-integrated soft robots.

In addition, we added supplementary experiments to compare signal stability

between the sensor in ref. [9] and PCAM sensor. As shown in added Supplementary Fig. 62, the sensor in ref. [9] showed worse long-term stability under robot actuation. The signal deviation (see definition in equation 6) reached 67% within one iteration of a 35-cm trajectory, while PCAM sensors illustrate <6% signal deviation among 5 iterations of a 60-cm trajectory (see Supplementary Fig. 34). The robustness of sensor signals greatly enhances the perception capabilities of integrated soft robots and plays key roles in machine learning process for predicting the robot trajectory. More stable signals lead to a more accurate trajectory prediction model. This part is detailly discussed in the response to comment 3.

Revision made

Page 15 of the revised manuscript and Page 68, 74, 75, 81, and 82 of Supporting Information.

Supplementary Table 6. Comparison among the sensing characteristics of PCAM sensor and ref. [9].

Sensing characteristics	Sensor in ref. [9] (Sci. Robot., 2019, 4, eaax7020)	PCAM Sensor (this work)
Gauge Factor (GF)	Maximal 1.3	Maximal 204
Linear Working Window	Fixed by robot structure. (0-60%)	Tunable from 5%-50%, to 10%-70% and 20%-120%
Sensing Performance Modelling	Not available	FEA tool

Supplementary Fig. 62. Digital photo and sensing profile of the origami robot in ref. [9] under a 35-cm trajectory. After calculation, the signal deviation reached 67% after completing one iteration of a 35-cm trajectory (moving forward mode).

Supplementary Fig. 34. Sensing profiles of a PCAM sensor-integrated pneumatic soft robot under trajectory 2, which was repeated 5 times. Red curves are sensor signals from right robot body and the black curves are sensor signals from left robot body.

Comment 3: The authors claimed the demonstration of a computational strain sensor design. But in fact, any good resistive strain sensor can be implemented in the algorithm.

Despite beautiful demonstration, from materials point of view, the novelty is not very high.

Response 3: Thanks Reviewer #2's comment. We agree that machine learning (ML) algorithm here is a general mathematical tool to analyze the sensing data, but not all resistive strain sensor can lead to an accurate prediction model, especially for the task of soft crawling robot autonomy in this work. Generally, accurate ML model requires high-quality sensing data to ensure the learning efficiency. In this work, strain sensors were attached on soft crawling robot to monitor their high degree-of-freedom body deformations and multi-modal actuation behaviors. For this task, instead of ML framework, critical challenges exist on how to obtain robust and stable sensing signals with high resolution during complex robot deformations/motions. From materials point of view, we are keen to highlight our contributions here are **(1) ultra-robust sensor design** and **(2) high-resolution sensor network on robot body** that greatly enhance ML learning efficiency and successfully realize the soft crawling robot autonomy.

Firstly, our PCAM sensor realized the excellent robustness that could work under complex operation environments. The importance of sensor stability has been highlighted by many review papers (*Adv. Mater.*, 2021, 33, 2004782; *Adv. Mater.*, 2019, 31, 1805921; *ACS Nano*, 2023, 17, 5211–5295.), but most sensors cannot sustain the noisy, intermittent, and dynamic environments that represent the complexity and uncertainty present in real-world working situations of soft robots (*Nat. Electron.*, 2022, 5, 784–793; *Nat. Commun.*, 2022, 13, 5311; *Adv. Mater.*, 2022, 34, 2203650; *Adv. Mater.*, 2019, 31, 1903789.). In this work, by utilizing programmed cracks array within micro-crumpled strategy, our developed PCAM sensors maintain robust sensing responses under various challenging operating conditions including noise interruptions (up to 50% strain), intermittent cyclic loadings (100,000 cycles), and dynamic operation frequencies (0-23 Hz). These robust sensors generated stable sensing data for diverse soft robots under complex motions, ensuring high ML learning efficiency when constructing the prediction model.

We added extra ML experiments to investigate the effects from sensor data stability on the accuracy of trajectory prediction model. As shown in Supplementary Figs. 33, 35, and 37, three types of sensors including PCAM sensor, crumpled sensor, and planar sensor were integrated on soft robot to perform a 25-cm turning trajectory for five times. Along with the increasing iteration, the signal deviations of PCAM sensor kept small (ca. 6%), but crumpled and planar sensors gradually increased to >50%. By using four iteration data as training set and adopting one iteration data as the testing set to construct the trajectory prediction model (see training and testing data files in GitHub). The model performance is evaluated by mean square error (MSE), where a lower MSE value indicates better model performance. As shown in

Supplementary Figs. 64 and 65, the PCAM sensor-based prediction mode showed three times lower MSE and more accurate trajectory prediction on the test set, illustrating the advantage of high learning efficiency from stable sensor data.

Besides, high-resolution sensor network provides another approach to improve the learning efficiency, which were realized by minimizing the number of sensors on the key locations of soft crawling robot body. For current ML applications on soft robots or actuators, the applied targets mainly refer to soft gloves (*Nat. Electron.*, 2021, 4, 193–201; *Nat. Electron.*, 2020, 3, 563–570; *Nat. Electron.*, 2020, 3, 571–578.) or soft grippers (*Sci. Robot.*, 2019, 4, eaax2198; *Sci. Robot.*, 2016, 1, eaai7529; *Proc. Natl. Acad. Sci. U.S.A.*, 2020, 117, 25352–25359; *Nat. Commun.*, 2020, 11, 5381.). To capture their motions, there is no doubt to attach sensors on all gripper or glove fingers. However, for the crawling origami robots in this work, there are >40 possibilities of the sensor locations. Therefore, we need to optimize the sensor number and location on the robot body to collect the most representative key information of robot motion. For example, in the added Supplementary Fig. 66, sensor at position 1 showed higher sensitivity than position 2 when tracking the robot motions. The simulation results in Supplementary Fig. 61 indicated that applying two sensors at origami head or tail is efficient for distinguish the turning direction. Therefore, to identify moving forward/backward and turning left/right motions, the minimized number and location of the sensors on robot was optimized as Supplementary Fig. 41. Besides, for the terrain prediction task, based on simulation results in Fig. 6a-6d, instead of four sensors at both robot head and tail, one sensor at head position is enough to distinguish the terrain height changes. These efforts simplified the relationship between sensor signals and robot motion to improve the learning efficiency. Therefore, simple ANN framework and <40 training samples were sufficient to generate the prediction model. To the best of our knowledge, as summarized in Supplementary Table 2, it is the first report to achieve robot autonomy on soft crawling robot (i.e., robotic trajectory prediction and topography altitude awareness).

We expect the above approaches provides new alternatives to enhance learning efficiency of ML and construct machine intelligence on more complex soft robot system.

Revision made

Page 15 and 16 of the revised manuscript and Page 70, 71, 72, 83, and 84 of Supporting Information.

Supplementary Fig. 35. Sensing performance of a PCAM sensor-integrated pneumatic soft robot under trajectory 1, which was repeated 5 times. (a) Digital photo of robot trajectory 1. **(b)** Comparison of signal deviations of pneumatic soft robots, which integrated PCAM sensor, planar sensor, and crumpled sensor, respectively. Herein, the signal deviations were calculated based on the sensing profiles based on the sensing profiles between 1st and 5th iterations under trajectory 1. Detailed calculation method was provided in Equation 6 in main manuscript.

Supplementary Fig. 33. Sensing profiles of a PCAM sensor-integrated pneumatic soft robot under trajectory 1, which was repeated 5 times. Red curves are sensor signals from right robot body and the black curves are sensor signals from left robot body.

Supplementary Fig. 37. Sensing profiles of a planar and crumpled sensor-integrated pneumatic soft robot under trajectory 1, which was repeated 5 times. (a) Sensing profile of a crumpled sensor-integrated pneumatic soft robot in the 1st iteration of trajectory 1. (b) Sensing profile of a crumpled sensor-integrated pneumatic soft robot in the 5th iteration of trajectory 1. (c) Sensing profile of a planar sensor-integrated pneumatic soft robot in the 1st iteration of trajectory 1. (d) Sensing profile of a planar sensor-integrated pneumatic soft robot in the 5th iteration of trajectory 1. Red curves are sensor signals from right robot body and the black curves are sensor signals from left robot body.

Supplementary Fig. 64. MSE of trajectory prediction model based on the training data from PCAM sensor, crumpled sensor, and planar sensor.

Supplementary Fig. 65. Prediction results of trajectory prediction model trained by data based on PCAM sensor, crumpled sensor, and planar sensor, respectively.

Supplementary Fig. 66. Sensor position optimization on origami robot. When monitoring the robot crawling motion, PCAM sensor at position 1 showed much higher signal response than position 2 thus is selected as the target position for constructing the body sensor network.

Supplementary Fig. 61. Simulation of origami deformation during turning. (a) Simulated movements of origami robot during moving forward and turning left. (b) Extracted folds distance changes between the designated two folds at both left and right sensor locations during moving forward and turning left.

Supplementary Fig. 41. Origami robot with four on-body sensors.

Fig. 6 Surrounding awareness robotic navigation. (a) Digital photo of robot navigation in an artificial terrain with varying altitudes. (b) Simulation of an origami robot climbing over a small hill. (c) Simulated folds distance changes at the origami robot head during the climbing and descending stages of climbing over a hill. (d) Sensing profiles of the sensor-integrated origami robot when climbing over six different

hills. Data smoothing by adjacent-averaging method, see details in Methods. (e) (f) Performance of the terrain prediction model on two test sets. (g) Comparison between model-predicted terrain altitudes and the ground truth recorded by video. The time reading is the record in Supplementary Video 10.

Comment 4: *Very recently there have been a number of reports showing soft materials with seal-healing actuation (e.g., <https://www.nature.com/articles/s41563-020-0736-2>), which is clearly more advanced than the system presented here. Eventually with cracks propagation after extensive cycles, the self-healing materials would outperform.*

Response 4: We understand that self-healing materials may help to connect cracks together after cycling, yet we are afraid that it may simultaneously bring some disadvantages of crack-based sensor, including **(1) sensor sensitivity**, **(2) working frequency**, and **(3) performance modelling**.

Firstly, self-healing materials will reduce the sensor sensitivity. Self-healing materials are usually non-conductive polymers. Compared with the pure SWNT material of PCAM sensors, the introduction of self-healing materials into the active SWNT layer will reduce the conductivity of sensing layer and lead to low sensor sensitivity (i.e., gauge factor (GF)). We added Supplementary Table 7 to compare the GF between this work and recent self-healing strain sensors.

Secondly, self-healing materials will limit sensor's working frequency. Most self-healing materials require a long healing time up to minutes or even hours. For the mentioned work of seal-healing actuation (*Nat. Mater.*, 2020, 19, 1230–1235), its healing time is greatly reduced to 1 second. Such a healing time means it can sustain a maximal working frequency of 1 Hz. Given that state-of-the-art soft actuators operate within a range of mutative speeds (0.1-20 Hz) (*Sci. Robot.*, 2019, 4, eaax1594; *Science*, 359, 2018, 61–65; *J. Am. Chem. Soc.*, 2021, 143, 4017–4023; *Sci. Adv.*, 2020, 6, eaaz6912.), it is essential to develop soft strain sensors to exhibit a stable sensing response across high operation frequencies, making them deployable for a broad range of soft robot applications. Our PCAM sensors have achieved this goal.

Thirdly but most importantly, self-healing materials will make the crack propagation behavior unpredictable, which is hardly to achieve sensor performance modelling. Self-healing materials usually experience a chemical bonding process between two interfaces, which is hard to simulate. On the other hand, once the cracks are healed, the stress-distribution map on the sensing layer will change from Supplementary Fig. 63a to 63b, and the route of crack growth and propagation cannot be regulated by the pre-determined crack array anymore and become uncontrollable. As a result, the sensing response curves cannot be modelled. In this work, we developed FEA tools that could simulate the structure evolution of PCAM sensor, capable of

predicting sensor’s sensing curves without conducting trial-and-error experiments. This strategy bridges the gap between physical modelling techniques and empirical experimentation, which is the first report to the best of our knowledge.

Considering the above issues of self-healing materials, we believe the PCAM design has its unique contribution to the crack-based strain sensors.

Revision made

Page 15 of the revised manuscript and Page 69, 75, 82, and 83 of Supporting Information.

Supplementary Table 7. Comparison among the GF of PCAM sensor and some recent self-healing strain sensors.

Reference	Self-healing materials	Maximal GF
F. Sun, et al. Nat. Commun. , 2023, 14, 130.	Yes	1.0
Y. Wang, et al. Adv. Funct. Mater. , 2023, 33, 2301587.	Yes	3.2
H. Fu, et al. Mater. Horiz. , 2022,9, 1412–1421.	Yes	2.2
D. Hardman, et al. NPG Asia Mater. , 2022, 14, 11.	Yes	1.5
T. Dai, et al. J. Mater. Chem. C , 2022,10, 15532–15540.	Yes	1.3
S. Liu, et al. ACS Appl. Polym. Mater. , 2020, 2, 3, 1325–1334.	Yes	0.14
G. Cai, et al. Adv. Sci. , 2017, 4, 1600190.	Yes	1.5
This work	No	204

Supplementary Fig. 63. Simulated stress distribution maps of PCAM sensors. (a) PCAM sensor under 50% uniaxial strains ($\phi=40\%$, $\rho=1,200 \mu\text{m mm}^{-2}$). **(b)** PCAM sensor with fully healed cracks under 50% uniaxial strains ($\phi=40\%$, $\rho=1,200 \mu\text{m mm}^{-2}$).

Comment 5: *Overall I consider the novelty of this work does not represent a milestone in this field and is not sufficient to warrant the publication in Nat. Comm.*

Response 5: We would like to appreciate Reviewer #2 for reviewing our manuscript in detail and giving us insightful comments. We have considered all the comments, suggestions, and concerns raised by Reviewer #2 and revised the manuscript point-by-point. The manuscript was improved a lot by adding extra supplementary experiments and discussions, including (1) highlight of the advantages and novelty of the developed PCAM sensor compared with ref. [9]; (2) discussion of our contributions on machine learning process from the material point of view; (3) comparison of the pros and cons of self-healing material and PCAM design on crack-based sensors. Overall, we believe that the story is much clear, and novelty is stronger. We hope that the revision is satisfactory to Reviewer #2.

Reviewer #3

Comment 1: *In this study, authors developed a sensor using the SWCNT-ecoflex composite by introducing Programmed cracks with an interlocking comb-shaped pattern. This sensor underwent heat treatment to achieve the desired shrinkage (i.e. pre-strain). The findings revealed that its performance, including sensitivity and linear working window, can be modulated using the crack density (ρ) and shrinkage ratio (ϕ). Both experimental results and Finite Element Method (FEM) analysis supported these findings. The team also put the sensor to the test under various conditions, such as dynamic mechanical loading, cyclic loading, and operation frequency tests, to ascertain its stability. All results indicated a robust stability. Furthermore, the versatility and robustness of this sensor were highlighted when it was successfully integrated into multiple soft robot applications. The team demonstrated its compatibility across a range of robotic scales: from large-scale origami robots to pneumatic robots and even tiny microrobots. In an intriguing application, the sensor, once attached to an origami robot, provided data that helped train an Artificial Neural Network (ANN) to predict the robot's movement and also to gauge the height of potential obstacles. The paper presents a set of multiple intriguing results, yet there's room for enhancement by considering the subsequent feedback.*

Response 1: We thank Reviewer #3 for his/her careful review and positive evaluation on our work. We have made point-by-point responses for each comment below.

Comment 2: *Figure 1e demonstrates the excellent reproducibility of the sensor. However, it only showcases scenarios with low crack density ($\rho=300\mu\text{m}/\text{mm}$). One might anticipate that as the crack density increases, the performance uncertainty may also rise. It would be beneficial to validate the reproducibility for higher crack densities. Alternatively, adding error bars to the results in Figure 1g for instances of higher crack density and discussing them would be advantageous.*

Response 2: Thank Reviewer #3's insightful suggestion. In the revised manuscript, we added error bars in Fig. 1g for PCAM sensors at higher crack density ($\rho=600$ and $1,200\mu\text{m}/\text{mm}^2$). Indeed, the sensor signal variations increased when ρ value increased from 300 to $1,200\mu\text{m}/\text{mm}^2$ with higher crack density. We also calculated the relative signal variation based on Equation S1,

$$\text{Relative Signal Variation} = \frac{S_{\delta_\varepsilon}}{\delta_\varepsilon} \quad (\text{S1})$$

, where δ_ε is the relative resistance change of PCAM sensor at ε strain, and S_{δ_ε} is the standard deviation of δ_ε from three sensor replicas at ε strain. According to Supplementary fig. 57, under different crack density (ρ) values, PCAM sensors showed

similar relative signal variations, which were <10% for most working windows. These results indicated a good performance reproducibility of the PCAM sensor.

Revision made

Page 7 and 26 of the revised manuscript and Page 63 and 77 of Supporting Information.

Fig. 1. Computation-guided PCAM sensor design. (a) Computer-aided design of interdigital crack pattern. (b) Illustration of pattern processing by a laser machine and optical images of the processed crack array. (c) SEM images of the fabricated strain sensor. (d) FEA simulated stress maps of PCAM sensor under strains. (e) Strain sensing profiles of a PCMA sensor ($\rho=300 \mu\text{m mm}^{-2}$, $\phi=40\%$) with three replicas. (f) Interdigital crack pattern with different crack densities (ρ values). (g) Strain sensing profile of PCAM sensors with different ρ values. The ϕ values of all sensors were kept as 40%. The error bars were calculated based on three replicas. (h) SEM images of PCAM sensors with different ϕ values. (i) Strain sensing profile of PCAM sensors with different ϕ values. The ρ values of all sensors were kept at $1,200 \mu\text{m mm}^{-2}$.

Supplementary Fig. 57. Relative signal variation of PCAM sensors based on three sensor replicas.

Comment 3: *In the Sensor design section on p.6, it's mentioned that sensitivity can be modulated by the crack density, and Figure 1g suggests that the linear working range remains fairly consistent across varying crack densities. However, it might appear that the linear working range expands as crack density increases. To prevent this potential misunderstanding, quantifying the linear working range might be a prudent approach.*

Response 3: We thank Reviewer #3's careful review and insightful suggestion. To avoid any misleading, we have quantified the linear working range of all PCAM sensors in the revised manuscript.

“When adopting crack arrays 1, 2, and 3 in Fig. 1f (see dimension details in Supplementary fig. 4), the ρ values were calculated as 300, 600, and 1,200 $\mu\text{m mm}^{-2}$, respectively. With increasing ρ values, as shown in Fig. 1g, PCAM sensors showed a little increased range of linear working windows from 17%-70% to 15-70% and 10%-70%, while their GF was greatly improved from 2.3 to 8.4 and 35.”

Revision made

Page 7 of the revised manuscript.

Comment 4: *Sensor performance can be assessed based on (i) sensitivity, (ii) linear working range, (iii) resilience under cyclic loading, and (iv) hysteresis during loading and unloading phases. While the authors have provided encouraging results for (i) through (iii), there's an absence of information regarding hysteresis. Incorporating hysteresis data would enhance the evaluation.*

Response 4: Thank Reviewer #3's comment. As shown in Supplementary Fig. 12, we have added the hysteresis data of PCAM sensor and made a discussion in the revised manuscript.

“In addition, the hysteresis of a PCAM sensor ($U_{hysteresis}$) was quantified by measuring the maximal signal difference between the stretching and releasing processes, as defined in Equation 5,

$$U_{hysteresis} = \text{Max}|\delta_{stretching} - \delta_{releasing}| \quad (5)$$

, where $\delta_{stretching}$ is the relative resistance change signal, $(R_{\epsilon}-R_0)/R_0$, of the PCAM sensor at ϵ strain during the stretching process, and $\delta_{releasing}$ is the relative resistance change of the PCAM sensor at ϵ strain during the relaxation process. Based on Supplementary Fig. 12, the hysteresis values of PCAM sensor under increasing applied strains (20%-70%) were calculated as 0.51, 1.75, 1.91, 2.90, 3.54, 3.63, respectively, as the hysteresis of the Ecoflex substrate increased with the applied strains (*Int. J. Mech. Sci.*, 2021, 206, 106624.).”

Revision made

Page 9 of the revised manuscript and Page 18 of Supporting Information.

Supplementary Fig. 12. Hysteresis profiles of a PCAM sensor ($\rho=1,200 \mu\text{m mm}^{-2}$; $\phi=40\%$) under uniaxial strain loading from 20% to 70%.

Comment 5: *Pertaining to point 3), the simulation was solely conducted for tensile loading and omitted the unloading phase. It would be valuable if the authors could incorporate a simulation illustrating the sensor's response during unloading. If this poses a challenge, a brief commentary on its practicability would be appropriate.*

Response 5: Thank Reviewer #3's comment and suggestion. We tried to conduct the performance modelling of sensor on the unloading process, yet current finite element analysis (FEA) model cannot accurately simulate the hysteresis effects of PCAM sensor. As discussed in the response to comment 4, there was a sensor signal gap between the

tensile loading and unloading phases due to the hysteresis of elastic sensor substrate. For the tensile loading process with an applied stretching force, the sensor mechanics quickly reach a steady-state thus FEA model could successfully simulate the sensor physical deformation and map the resistance change. However, during the unloading phase, the mechanical recovery of sensor is mainly affected by the spontaneously stress relaxation of elastic polymer segments of sensor substrate, which is a nanoscale physical process and shows non-steady feature. Such microprocess and non-steady state are hard to be simulated by the macroscopic and steady-state suited FEA model (the minimal element size of FEA model in this work was $150\ \mu\text{m}\times 150\ \mu\text{m}\times 100\ \mu\text{m}$). We also tried to search potential design ideas from literature. Although there were some FEA models to simulate the mechanical sensor deformations under applied strains (*Nat. Mach. Intell.*, 2022, 4, 84–94; *Nat. Commun.*, 2022, 13, 6604; *Adv. Funct. Mater.*, 2008, 18, 3553–3567.), the sensing responses at strain unloading phase remains an unsolved problem. We will conduct in-depth studies on this aspect at next research stage.

Revision made

Page 8 of the revised manuscript.

Comment 6: In the 'Intelligent sensor network for robotic trajectory prediction' section, the authors reference the use of actuation information and sensing information for ANN training. Delving into the specifics of what comprises the actuation information would be insightful.

Response 6: We thank Reviewer #3's comment and suggestion. We have added the detail contents of actuation information in the revised manuscript.

“These sensor data combined with robot actuation information, which served as model inputs. Herein, as shown in Fig. 5a, the actuation information contained the instructions from robot control station including crawling direction (1 refers to crawling forward, -1 refers to crawling backward, and 0 refers to no crawling motion) and turning direction (1 refers to turning left, -1 refers to turning right, and 0 refers to no turning motion).”

Revision made

Page 12 of the revised manuscript.

Comment 7: Moreover, both this section and the 'Surrounding awareness robot navigation' section indicate the use of 38 and 30 training data points respectively. An explanation of the methodology behind generating the initial training data would enhance understanding.

Response 7: Thank Reviewer #3's suggestion. We have added new sections in the

Methods to supplement the details of the methodology to generate the initial training data.

“Dataset collection for robotic trajectory prediction model. First, training data consisted of independent and dependent variables. In this work, independent variables contained multi-channelled sensor data as well as robot actuation information, while the dependent variable was the robot trajectory (i.e., real-time robot location in the terrain). Herein, the sensor data was recorded by the multimeter Keithley DMM6500, and robot actuation information was extracted from robot control station, while robot trajectory was recorded a camera system (see Supplementary fig. 44) which was further extracted using a “Tracker” program. With the setup in Fig. 5b, the sensor-integrated origami robot was placed in an artificial terrain to execute different trajectories, including 7 modes: (1) moving forward N steps; (2) moving forward N steps then meeting an obstacle with N steps; (3) moving forward N steps then turning left N steps (meeting an obstacle with N steps or not); (4) moving forward N steps then turning right N steps (meeting an obstacle with N steps or not); (5) moving forward N steps then moving backward N steps; (6) moving forward N steps then moving backward N steps then turning left N steps (meeting an obstacle with N steps or not); (7) moving forward N steps then moving backward N steps then turning right N steps (meeting an obstacle with N steps or not). By randomly setting different N values, we collected 43 datasets which were not repeated. As shown in Fig. 5d, we selected 5 datasets with representative trajectories as testing datasets and took other 38 datasets as training datasets.

Dataset collection for terrain prediction model. First, training data consisted of independent and dependent variables. In this work, independent variable was on-body sensor data, and the dependent variables were the robotic trajectory (i.e., real-time robot location) as well as the terrain information (i.e., terrain altitude changes along the robotic trajectory). Herein, the sensor data was recorded by the multimeter Keithley DMM6500, and robot trajectory was recorded a camera system (see Supplementary fig. 44) which was further extracted using a “Tracker” program, and the terrain consisted of 6 different small hills (2 hills with 1.5mm height, 2 hills with 3.0mm height, 2 hills with 4.5mm height) and their distribution information was recorded by the researchers. With the setup in Fig. 5a, a sensor-integrated origami robot was placed in the artificial terrain to climb over 6 hills with different combinations, such as 1.5-3.0-1.5-4.5-3.0-4.5, 4.5-3.0-1.5-4.5-3.0-1.5, and so on. By randomly setting different hill combinations, we collected 32 datasets which were not repeated. As shown in Fig. 6e, we selected 2 datasets with representative hill distributions as testing datasets and took other 30 datasets as training datasets.”

Revision made

Page 21 and 22 of the revised manuscript.

Comment 8: *Also, the decision to use a 10-fold validation on a dataset comprising roughly 30 samples may seem disproportionate, leading one to question if the selected k-value is overly high for the dataset size.*

Response 8: Thank Reviewer #3's comment. We did extra experiments and tried different k_fold values from 4 to 10 to validate the effects on the resulting ML model. The model performance is evaluated by mean square error (MSE), where a lower MSE value indicates better model performance. As summarized in Supplementary Table 9, when using different k_fold values, the MSE of the prediction model on both validation and test sets were very similar, and the model with 10_fold validation showed the lowest MSE and thus was picked as the model parameter.

Revision made

Page 22 of the revised manuscript and Page 76 and 84 of Supporting Information.

Supplementary Table 9. Performance of robotic trajectory prediction model with different k_fold values.

Model Parameter			Train	Validation	Test
Epoch	Bath size	k_fold	MSE	MSE	MSE
10	10	4	10.82	9.14	4.67
10	10	6	8.96	8.45	3.99
10	10	8	8.33	8.13	4.06
10	10	10	9.17	8.13	3.97

Comment 9: *Considering the fabrication procedure, it appears that the electrically percolating SWCNT layer might not be completely encased within the ecoflex matrices. A sandwiched structure could potentially decrease hysteresis and improve wear resistance. Including a discussion on this perspective would be valuable.*

Response 9: We agree with Reviewer #3 that, by coating a thin Ecoflex layer on the sensor top surface, a sandwiched structure of sensor could decrease hysteresis and improve wear resistance, which is desired in practical applications. However, on the other hand, an additional Ecoflex layer on the top surface also increase the complexity in sensor performance modelling, and the encapsulation of wire electrodes inside the sensor also need a proper design. Therefore, the consideration of performance balance is required. We have supplemented above discussion in the revised manuscript.

Revision made

Page 9 of the revised manuscript and Page 78 of Supporting Information.

Comment 10: *Concerning the underlying mechanism of the 'delta R-strain' response, the authors suggest a gradual appearance of the surface crack with the increase of strain. However, I anticipate that the entire surface crack path would open abruptly at the applied strain where the compressive pre-strain from initial thermal contraction is nullified. Thus, from zero strain to this critical strain point, the resistance would see a gradual rise (as seen from 0-20% strain in Fig 1e). Once the surface crack emerges, a stiffer resistance increase would ensue due to the elongated serpentine-like electrical path (as observed from 20-60% strain in Fig 1e). Introducing a stress-strain curve (from experiments and simulation) in the discussion might offer more clarity in testing this perspective.*

Response 10: We thank Reviewer #3's insightful comment and suggestion. As the SWNT layer with the interdigital crack array cannot be freestanding, it is hard to test its stress-strain curve under in-plane stretching. Instead, we did more *in situ* scanning electron microscope (SEM) studies and finite element analysis (FEA) simulation to reveal the structure evolution and mechanism of PCAM sensor in Fig. 1e. The extra experimental results and discussion was supplemented in the revised manuscript as below.

“Fig. 1e presents the strain sensing profiles of a PCAM sensor with a simple interdigital pattern 1 (see dimension details in Supplementary fig. 4b), showing a three-stage sensing response. First, there is a silent region from 0 to ca. 20% strains, where the resistance changes were small. Afterward, it experienced a quick response from 20% to ca. 60% and then gradually reached a plateau. We implemented *in situ* SEM studies and FEA simulation to reveal the structure evolutions and mechanism. As shown in Supplementary Fig. 7, there was relative uniform stress distribution across the SWNT sensing layer when applying <20% strain, and the in-plane micro-crumpled were gradually deformed to release the compressive pre-strains from thermal contraction during sensor fabrication, and slowly widening the cracks. When the applied strain $\geq 20\%$, low stress was concentrated on the crack trace locations, which were more easily deformed, and the cracks started growing quickly until ca.60% strain. Thereafter, the length of each crack reached its maximum value, and further cracks grew in the width instead of the length, minimally affecting the conductive pathways within the SWNT layer of sensor. As a result, the sensing signal changes became milder at this stage.”

Revision made

Page 5 and 6 of the revised manuscript and Page 13 of Supporting Information.

Supplementary Fig. 7. FEA simulation (a) and structure evolution (b) of a PCAM sensor under uniaxial strain loading. ($\rho=300 \mu\text{m mm}^{-2}$; $\varphi=40\%$).

REVIEWERS' COMMENTS

Reviewer #2 (Remarks to the Author):

The authors addressed my questions reasonably well, as reflected by the detailed comparison with prior arts and differentiation. Nevertheless, I still find the study represents a rather incremental improvement compared to the literature. I have no question about the research efforts devoted to this study, but the novelty, in particular about the materials design and sensing mechanism, is really not sufficient to me.

Reviewer #3 (Remarks to the Author):

The authors have effectively responded to the reviewer's comments, enhancing both the rigor and clarity of their paper. I recommend its publication in its current state.

Responses to the Reviewers' Comments

Reviewer #2

Comment 1: *The authors addressed my questions reasonably well, as reflected by the detailed comparison with prior arts and differentiation. Nevertheless, I still find the study represents a rather incremental improvement compared to the literature. I have no question about the research efforts devoted to this study, but the novelty, in particular about the materials design and sensing mechanism, is really not sufficient to me.*

Response 1: We thank Reviewer #2 for his/her satisfaction on our manuscript revision. As for the novelty, we believe our contributions here are (1) accurate sensor performance modelling, (2) excellent sensor signal robustness, and (3) high-level soft robot integration, rather than the materials and sensing mechanism where we used commercial carbon nanotube materials and general crack propagation mechanism.

Reviewer #3

Comment 1: *The authors have effectively responded to the reviewer's comments, enhancing both the rigor and clarity of their paper. I recommend its publication in its current state.*

Response 1: We thank Reviewer #3 for his/her careful review and valuable suggestions, the manuscript is improved a lot after revision.